# CONFORMAL PREDICTION FOR LONG-TAILED CLASSIFICATION

**Tiffany Ding**[1]    **Jean-Baptiste Fermanian**[2]    **Joseph Salmon**[2]
[1]University of California, Berkeley, [2]University of Montpellier, Inria, CNRS
`tiffany_ding@berkeley.edu`
`{jean-baptiste.fermanian, joseph.salmon}@inria.fr`

## ABSTRACT

Many real-world classification problems, such as plant identification, have extremely long-tailed class distributions. In order for prediction sets to be useful in such settings, they should *(i) provide good class-conditional coverage*, ensuring that rare classes are not systematically omitted from the prediction sets, and *(ii) be a reasonable size*, allowing users to easily verify candidate labels. Unfortunately, existing conformal prediction methods, when applied to the long-tailed setting, force practitioners to make a binary choice between small sets with poor class-conditional coverage or sets that have very good class-conditional coverage but are extremely large. We propose methods with marginal coverage guarantees that smoothly trade off set size and class-conditional coverage. First, we introduce a new conformal score function called prevalence-adjusted softmax that optimizes for macro-coverage, defined as the average class-conditional coverage across classes. Second, we propose a new procedure that interpolates between marginal and class-conditional conformal prediction by linearly interpolating their conformal score thresholds. We demonstrate our methods on Pl@ntNet-300K and iNaturalist-2018, two long-tailed image datasets with 1,081 and 8,142 classes, respectively.

## 1 INTRODUCTION

Prediction sets are useful because they replace a single fallible point prediction with a set that is likely to contain the true label. In classification, prediction sets are most useful in settings *with many classes*, as they narrow down the label space to a set of labels that human decision makers can then verify. Consider an amateur plant enthusiast who wants to identify a plant. The enthusiast struggles to identify plants on their own, but when presented with a short list of possible matches, it is easy for them to go through the list and select the correct species (e.g., by comparing their plant with images of potential matches).

Another key characteristic of plant identification is its *extremely long-tailed class distribution*. As shown in Figure 1, there are thousands of images of common plants but only a handful for rare plants. Such skewed distributions also appear in animal identification and disease diagnosis. An added challenge is that we often care even more about identifying instances of the rare classes than the popular ones. In botany, scientists may want to prioritize the acquisition of examples of endangered plant species, which fall in the tail (Figure 1). In medicine, catching the few cases of an aggressive cancer early matters more than perfectly classifying common benign lesions. In collaborative human-AI systems where a human generates labels based on AI recommendations and these labels are used to improve the predictive model in future training rounds, neglecting niche classes can accelerate "model collapse" (Shumailov et al., 2024), shrinking the model's effective label space over time and degrading accuracy. This motivates pursuing prediction sets where all classes have a high probability of being correctly included in the prediction set. Beyond training good predictive models in this setting, an additional challenge for post-hoc uncertainty quantification is that most available examples of rare classes are used for model training, leaving these classes with few or zero holdout examples to use for calibrating uncertainty quantification methods.

Formally, let $X \in \mathcal{X}$ be features with unknown label $Y \in \mathcal{Y} = \{1, 2, \ldots, |\mathcal{Y}|\}$. Our goal is to construct a set-generating procedure $\mathcal{C} : \mathcal{X} \rightarrow 2^{\mathcal{Y}}$, where $2^{\mathcal{Y}}$ denotes the set of all subsets of $\mathcal{Y}$, with good class-conditional coverage. For $y \in \mathcal{Y}$, the *class-conditional coverage of $\mathcal{C}$ for class $y$*

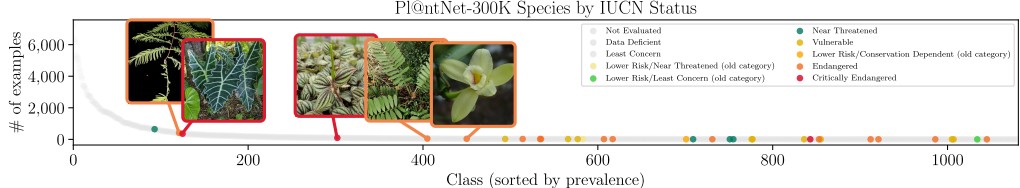

Figure 1: The number of `train` examples of each species in Pl@ntNet-300K (Garcin et al., 2021). We highlight threatened species, as defined by the International Union for Conservation of Nature (https://iucn.org), which are particularly important to identify for biodiversity monitoring purposes. Note that most of these species are in the tail of the distribution.

is $\text{CondCov}(\mathcal{C}, y) = \mathbb{P}(Y \in \mathcal{C}(X) \mid Y = y)$. This can be contrasted with *marginal coverage*, which is simply $\text{MarginalCov}(\mathcal{C}) = \mathbb{P}(Y \in \mathcal{C}(X))$. To ensure that our prediction sets are useful for identifying instances of all classes, including rare ones, we aim for high class-conditional coverage across all classes. In addition to coverage, set size is also crucial, as large sets are often impractical. For instance, in plant identification, users lack the time to review prediction sets containing hundreds of species. Ideally, we would generate prediction sets with both high class-conditional coverage and small size. Unfortunately, there is an inherent trade-off between small set sizes and class-conditional coverage in the long-tailed setting.

Conformal prediction (CP) provides methods guaranteed to achieve marginal or class-conditional coverage under no distributional assumptions. STANDARD CP, the most basic conformal prediction method, yields small sets but only guarantees marginal coverage and often has poor class-conditional coverage for some classes. However, approaches targeting class-conditional coverage struggle in scenarios where many classes have few examples. In such settings, CLASSWISE CP (an instantiation of Mondrian conformal; see Vovk et al., 2005) and rank-calibrated class-conditional CP (Shi et al., 2024) produce large sets, and Clustered CP (Ding et al., 2023) effectively defaults to STANDARD CP for rare classes that it is unable to assign to a cluster.

**Objective.** Our goal is to produce prediction sets that maintain marginal coverage while striking a more useful trade-off between set size and class conditional coverage compared to STANDARD or CLASSWISE CP. We approach this in two ways.

APPROACH I: *Target a relaxed notion of class-conditional coverage.* Motivated by the multi-class classification concept of "macro-accuracy," which is the average of class-wise accuracies (Lewis, 1991), we aim to construct sets with high ***macro-coverage***, which is the average of class-conditional coverages:

$$\text{MacroCov}(\mathcal{C}) = \frac{1}{|\mathcal{Y}|} \sum_{y \in \mathcal{Y}} \mathbb{P}(Y \in \mathcal{C}(X) \mid Y = y) = \frac{1}{|\mathcal{Y}|} \sum_{y \in \mathcal{Y}} \text{CondCov}(\mathcal{C}, y). \quad (1)$$

In contrast, marginal coverage is a weighted average of class-conditional coverages where the weight of class $y$ is its prevalence $p(y)$:

$$\text{MarginalCov}(\mathcal{C}) = \sum_{y \in \mathcal{Y}} p(y) \cdot \text{CondCov}(\mathcal{C}, y). \quad (2)$$

This (over)emphasizes coverage of more frequent classes. We derive the form of the prediction set that optimally trades off set size and macro-coverage under oracle knowledge of the underlying distribution and design a conformal score function, called **prevalence-adjusted softmax** (PAS), which approximates these oracle optimal sets given an imperfect classifier $\hat{p}(y|x)$ and estimated label distribution $\hat{p}(y)$.

APPROACH II: *Target class-conditional coverage, then back off (until the set size is reasonable).* We propose a simple procedure called **INTERP-Q** that interpolates between CLASSWISE CP and STANDARD CP in a literal sense by linearly interpolating their quantile thresholds. This method allows the user to choose their position on the trade-off curve between set size and class-conditional coverage via the interpolation parameter.

The choice between these two approaches depends on the setting. Targeting macro-coverage (APPROACH I) implies that we care equally about the coverage of all classes, but it is acceptable

if a few classes have poor coverage, so long as the average class-conditional coverage is high. On the other hand, starting from CLASSWISE and softening (APPROACH II) implies that we want *all* classes to have good coverage. This approach also comes with a parameter that the user can vary depending on their preference between small sets and class-conditional coverage.

## 1.1 RELATED WORK

**Class-conditional conformal prediction.** Conformal prediction provides a way to construct prediction sets with coverage guarantees given a calibration dataset that is exchangeable with the test point (Vovk et al., 2005; Angelopoulos & Bates, 2023). We are specifically interested in class-conditional coverage, which is difficult to achieve in a useful way when many classes have very few calibration examples, as is the case in long-tailed settings. Previous works on class-conditional conformal focus on easy settings with at most ten classes (Shi et al., 2013; Löfström et al., 2015; Hechtlinger et al., 2018; Sadinle et al., 2019), or the harder setting of many classes with some class imbalance but still lacking a truly long tail (Ding et al., 2023; Shi et al., 2024).

**Optimally trading off set size and coverage.** Prediction sets should achieve the desired coverage while being as small as possible, so as to be maximally informative. In general, the set-generating procedures that optimally navigate the coverage–size trade-off depend on the density of the underlying distribution (Lei et al., 2013; Lei & Wasserman, 2014; Vovk et al., 2016; Sadinle et al., 2019). Although the true density is unknown in practice, these theoretically optimal sets serve as guidelines for designing effective conformal score functions or new conformal procedures for various target quantities, such as $X$-conditional coverage (e.g., APS from Romano et al., 2020, RAPS from Angelopoulos et al., 2021, SAPS from Huang et al., 2023, CQC from Cauchois et al., 2021) or size (Denis & Hebiri, 2017; Kiyani et al., 2024). In APPROACH I, we construct a score function inspired by the oracle sets that optimally trade-off set size and *macro-coverage*, a coverage target that has not been previously explored in this context.

**Learning from long-tailed data.** Many real-world classification problems exhibit long-tailed distributions, a challenge particularly pronounced in *fine-grained visual categorization*, a field that has recently attracted substantial interest, especially in biodiversity (Van Horn et al., 2015; 2018; Garcin et al., 2021; Wang et al., 2022). The goal is to classify images into highly specific subcategories (e.g., plant species), which often differ only subtly. While extensive research has focused on improving class-conditional top-1 or top-$k$ accuracy for such tasks (Russakovsky et al., 2015; Lapin et al., 2015; Liu et al., 2019; Garcin et al., 2021; Zhang et al., 2023), less attention has been devoted to constructing high-quality prediction sets in long-tailed settings, beyond the naive approach of selecting the top-$k$ predictions. While methods like logit adjustment for long-tail learning (Menon et al., 2021) and focal loss for dense object detection (Lin et al., 2017) address class imbalance through reweighting or output adjustment to improve classification performance, designing robust prediction sets under long-tailed distributions remains understudied. Addressing this gap is critical for systems like Pl@ntNet (Joly et al., 2014), a plant identification app where users upload images and receive candidate species matches.

## 1.2 PRELIMINARIES

**Notation.** For a positive integer $n$, let $[n] := \{1, \ldots, n\}$. Let $\mathcal{D}_{\text{cal}} = \{(X_i, Y_i) \text{ for } i \in [n]\}$ be a calibration set, where $(X_i, Y_i)$ for $i = 1, \ldots, n$ are exchangeable with the test point $(X_{n+1}, Y_{n+1})$, where the label $Y_{n+1}$ is unknown. For a class $y \in \mathcal{Y}$, we use $\mathcal{I}_y = \{i \in [n] : Y_i = y\}$ to denote the set of calibration points with label $y$ and $n_y = |\mathcal{I}_y|$ the number of calibration points with label $y$. Let $\alpha \in [0, 1]$ be a user-specified probability of miscoverage. We use $s : \mathcal{X} \times \mathcal{Y} \to \mathbb{R}$ to denote a conformal score function, where smaller values of $s(x, y)$ indicate that the pair $(x, y)$ conforms better with previously seen data.

**Score function.** We focus on the split conformal setting in which the score function $s$ is constructed using a predictive model trained on data separate from the calibration set $\mathcal{D}_{\text{cal}}$. We use the softmax conformal score function, defined as $s_{\text{softmax}}(x, y) = 1 - \hat{p}(y|x)$, where $\hat{p}(y|x)$ is the predicted probability for class $y$ for input $x$ obtained from the softmax output of a trained neural network. This is also known as the Least Ambiguous Classifier (LAC) score (Sadinle et al., 2019). Our methods can be readily adapted to other scoring functions but we focus on this score because alternatives like APS (Romano et al., 2020) and its variants produce larger sets and are primarily designed for $X$-conditional coverage, which is outside our scope.

---

**Algorithm 1** Conformal prediction

---

**Require:** Score function $s : \mathcal{X} \times \mathcal{Y} \to \mathbb{R}$, miscoverage level $\alpha \in [0, 1]$, calibration set $\mathcal{D}_{\text{cal}} = \{(X_i, Y_i)\}_{i=1}^n$, test point $X_{n+1}$, threshold function $\hat{\mathbf{q}} : \mathbb{R}^n \times [0, 1] \to \mathbb{R}^{|\mathcal{Y}|}$

 Compute scores: $S_i \leftarrow s(X_i, Y_i)$ for $i = 1, \ldots, n$

 Compute thresholds: $\mathbf{q} \leftarrow \hat{\mathbf{q}}((S_i)_{i=1}^n, \alpha)$

 **return** Prediction set $\mathcal{C}(X_{n+1}) = \{y : s(X_{n+1}, y) \leq q_y\}$, where $q_y$ is the $y$-th entry of $\mathbf{q}$

---

**Conformal prediction as thresholded sets.** Let $\mathbf{q} = (q_1, q_2, \ldots, q_{|\mathcal{Y}|})$ be a $|\mathcal{Y}|$-dimensional vector of score thresholds. Define the $\mathbf{q}$-*thresholded set* as

$$\mathcal{C}(X; \mathbf{q}) = \{y \in \mathcal{Y} : s(X, y) \leq q_y\}. \tag{3}$$

Conformal prediction provides principled ways to set $\mathbf{q}$ as a function of the calibration data $\mathcal{D}_{\text{cal}}$ and the chosen miscoverage level $\alpha$ so as to achieve coverage guarantees.

STANDARD *conformal prediction* constructs sets as $\mathcal{C}_{\text{STAND.}}(X) := \mathcal{C}(X; \hat{\mathbf{q}}_{\text{STAND.}})$ for $\hat{\mathbf{q}}_{\text{STAND.}} = (\hat{q}, \ldots, \hat{q})$ where

$$\hat{q} = \text{Quantile}_{1-\alpha}\Big(\frac{1}{n+1}\sum_{i=1}^n \delta_{s(X_i, Y_i)} + \frac{1}{n+1}\delta_\infty\Big), \tag{4}$$

and where $\text{Quantile}_\gamma(P) = \inf\{x \in \mathbb{R} : \mathbb{P}(V \leq x) \geq \gamma\}$ denotes the level-$\gamma$ quantile of a random variable $V \sim P$ and $\delta_s$ is the Dirac measure at point $s$ (this follows the notation of Tibshirani et al., 2019). By setting the thresholds in this way, STANDARD conformal prediction sets achieve a *marginal coverage* guarantee (Vovk et al., 2005):

$$\mathbb{P}(Y \in \mathcal{C}(X; \hat{\mathbf{q}}_{\text{STAND.}})) \geq 1 - \alpha. \tag{5}$$

CLASSWISE *conformal prediction* constructs sets as $\mathcal{C}_{\text{CLASSWISE}}(X) := \mathcal{C}(X; \hat{\mathbf{q}}_{\text{CLASSWISE}})$ where the $y$-th entry of $\hat{\mathbf{q}}_{\text{CLASSWISE}}$ is

$$\hat{q}_y^{\text{CW}} = \text{Quantile}_{1-\alpha}\Big(\frac{1}{n_y + 1}\sum_{i \in \mathcal{I}_y} \delta_{s(X_i, Y_i)} + \frac{1}{n_y + 1}\delta_\infty\Big). \tag{6}$$

CLASSWISE conformal prediction sets achieve a *class-conditional coverage* guarantee (Vovk et al., 2005):

$$\mathbb{P}(Y \in \mathcal{C}(X; \hat{\mathbf{q}}_{\text{CLASSWISE}}) \mid Y = y) \geq 1 - \alpha, \qquad \text{for all } y \in \mathcal{Y}. \tag{7}$$

By marginalizing over $y$, this implies that $\mathcal{C}_{\text{CLASSWISE}}(X)$ also achieves $1 - \alpha$ marginal coverage.

We explicitly describe the meta-algorithm for conformal prediction in Algorithm 1. The existing methods, STANDARD and CLASSWISE, and the methods we will propose in the next section are instantiations of this meta-algorithm for the score functions and threshold functions described in Table 1.

## 2 METHODS

We take two approaches to the problem of simultaneously achieving reasonable set sizes and reasonable class-conditional coverage in the long-tailed classification setting, each leading to a method. APPROACH I targets the weaker objective of macro-coverage. From this, we derive an oracle set that optimally balances set size and macro-coverage. We then define a conformal score function (PAS) by replacing the true conditional density with its estimate. We also consider an extension (WPAS) that prioritizes coverage of user-specified classes. APPROACH II addresses the trade-off by interpolating between class-

Table 1: Summary of the conformal methods considered. MargCov refers to the marginal coverage guarantee of the method.

|  | Score function | Threshold function | MargCov |
|---|---|---|---|
| STANDARD | any | $\hat{\mathbf{q}}_{\text{STAND.}}$ (4) | $1 - \alpha$ |
| CLASSWISE | any | $\hat{\mathbf{q}}_{\text{C.WISE}}$ (6) | $1 - \alpha$ |
| PAS* | $s_{\text{PAS}}$ (11) | $\hat{\mathbf{q}}_{\text{STAND.}}$ (4) | $1 - \alpha$ |
| WPAS* | $s_{\text{WPAS}}$ (14) | $\hat{\mathbf{q}}_{\text{STAND.}}$ (4) | $1 - \alpha$ |
| INTERP-Q* | any | $\hat{\mathbf{q}}^{\text{IQ}}$ (15) | $1 - 2\alpha$ |

*Our methods

conditional and marginal score quantiles, leading to a simple new procedure (INTERP-Q). We defer the formal statements of propositions and their proofs to Appendix B.

## 2.1 Approach I: Targeting (weighted) macro-coverage via a new score

We consider the population optimization problem of minimizing the expected set size subject to a macro-coverage constraint,

$$\min_{\mathcal{C}:\mathcal{X}\mapsto 2^{\mathcal{Y}}} \mathbb{E}[|\mathcal{C}(X)|] \quad \text{subject to } \mathrm{MacroCov}(\mathcal{C}) \geq \beta, \tag{8}$$

and its dual version maximizing macro-coverage subject to an expected set size constraint,

$$\max_{\mathcal{C}:\mathcal{X}\mapsto 2^{\mathcal{Y}}} \mathrm{MacroCov}(\mathcal{C}) \quad \text{subject to } \mathbb{E}[|\mathcal{C}(X)|] \leq \kappa \tag{9}$$

where $\mathrm{MacroCov}(\mathcal{C})$ is defined in (1) and $\beta \geq 0$ and $\kappa \geq 0$.

**Proposition 1** (Informal). *The solutions of (8) and (9) are of the form*

$$\mathcal{C}^*(x) = \{y \in \mathcal{Y} : p(y|x)/p(y) \geq t\}, \tag{10}$$

*for some threshold $t$ that depends on $\beta$ or $\kappa$, respectively.*

The key takeaway from this proposition is that thresholding on $p(y|x)/p(y)$ optimally balances macro-coverage and expected set size. Specifically, among set-generating procedures with a given expected set size, none achieve better macro-coverage than thresholding on $p(y|x)/p(y)$. Similarly, for a fixed macro-coverage, no other procedure yields a smaller expected set size. We contrast this with the solution to the more classical problem of minimizing expected set size subject to marginal or class-conditional coverage, which is given by thresholding on $p(y|x)$ (Neyman & Pearson, 1933; Sadinle et al., 2019).

Although we do not have access to $p(y|x)$ and $p(y)$ in practice, we have estimates $\hat{p}(y|x)$ and $\hat{p}(y)$ from our classifier and the distribution of empirical training labels, respectively. By creating prediction sets as $\widehat{\mathcal{C}}(x) = \{y \in \mathcal{Y} : \hat{p}(y|x)/\hat{p}(y) \geq t\}$ for a threshold $t$, we approximate an oracle Pareto-optimal set. We choose $t$ to achieve $1 - \alpha$ marginal coverage in the following way: Observe that $\widehat{\mathcal{C}}$ can be rewritten as $\widehat{\mathcal{C}}(x) = \{y \in \mathcal{Y} : s_{\mathsf{PAS}}(x,y) \leq -t\}$, where

$$s_{\mathsf{PAS}}(x,y) = -\hat{p}(y|x)/\hat{p}(y) \tag{11}$$

and PAS stands for *prevalence-adjusted softmax*. By setting $-t$ as the Standard CP $\hat{q}$ from (4) using $s_{\mathsf{PAS}}$ as the score, $\widehat{\mathcal{C}}$ inherits the marginal coverage guarantee of Standard CP. In summary, the first method we propose is simply running Standard CP with the PAS score function (which we will refer to as Standard with PAS), as this achieves the desired marginal coverage guarantee while (approximately) optimally trading off set size and macro-coverage. We emphasize that PAS aims to better handle this trade-off but does not directly target a macro-coverage guarantee.

**Extension to weighted macro-coverage.** Recall that macro-coverage is the unweighted average of the class-conditional coverages, and the PAS score function is designed to optimize macro-coverage among all set-generating procedures with a certain expected set size. However, in some settings, we may instead wish to optimize for a *weighted* average of the class-conditional coverages (e.g., because it is more important to cover some classes than others). Given user-chosen class weights $\omega(y)$ for $y \in \mathcal{Y}$ that sum to one, we can similarly define the $\omega$-*weighted macro-coverage* as

$$\mathrm{MacroCov}_{\omega}(\mathcal{C}) = \sum_{y \in \mathcal{Y}} \omega(y)\mathbb{P}(Y \in \mathcal{C}(X) \mid Y = y). \tag{12}$$

For $\omega(y) = |\mathcal{Y}|^{-1}$ we recover MacroCov and for $\omega(y) = p(y)$ we get MarginalCov.

**Proposition 2** (Informal). *The solutions of (8) and (9) when MacroCov is replaced with MacroCov$_{\omega}$ are of the form*

$$\mathcal{C}^*(x) = \{y \in \mathcal{Y} : \omega(y) \cdot p(y|x)/p(y) \geq t\}, \tag{13}$$

*for some threshold $t$ that depends on $\omega$ and $\beta$ or $\kappa$, respectively.*

We can approximate these optimal sets by running Standard CP with the *weighted prevalence-adjusted softmax* (WPAS),

$$s_{\mathsf{WPAS}}(x,y) := -\omega(y)\frac{\hat{p}(y|x)}{\hat{p}(y)}, \tag{14}$$

as the score function.

## 2.2 Approach II: Softening Classwise CP via Interp-Q

In the previous section, we presented our first solution, a new conformal score function; here we present our second solution, which is a simple way to interpolate between the behaviors of Classwise and Standard by linearly interpolating their quantile thresholds. We call this procedure Interp-Q for "interpolated quantile" because it constructs sets as $\mathcal{C}_{\text{INTERP-Q}}(X) := \mathcal{C}(X; \hat{\mathbf{q}}_{\text{IQ}})$ where the $y$-th entry of $\hat{\mathbf{q}}_{\text{IQ}}$ is a weighted average of $\hat{q}$ and $\hat{q}_y^{\text{CW}}$, as defined in (4) and (6), for some weight $\tau \in [0, 1]$. That is,

$$\hat{q}_y^{\text{IQ}} = \tau \hat{q}_y^{\text{CW}} + (1 - \tau)\hat{q} \qquad \text{for all } y \in \mathcal{Y}. \tag{15}$$

For classes where $\hat{q}_y^{\text{CW}} = \infty$ due to small $n_y$, we replace it with one (the maximum possible value of the $s_{\text{softmax}}$ conformal score) before interpolating.

**Proposition 3.** *If $\hat{q}$ and $\hat{q}_y^{\text{CW}}$ (for $y \in \mathcal{Y}$) are the* Standard *and* Classwise *conformal quantiles for $\alpha \in [0, 1]$, then $\mathcal{C}_{\text{INTERP-Q}}$ achieves a marginal coverage of at least $1 - 2\alpha$.*

Theoretically, this lower bound is almost tight; we can construct a pathological example where Interp-Q achieves coverage of $1 - 2\alpha + \alpha^2$, which is close to $1 - 2\alpha$ for small $\alpha$ (see Appendix B.3). Empirically, however, we find that Interp-Q achieves coverage close to $1 - \alpha$. This is because the real data we test on do not exhibit the pathologies from our example (which uses discrete score distributions that differ greatly between classes).

In Appendix A, we consider an alternative way of instantiating the interpolation idea used by Interp-Q via a method we call Fuzzy Classwise CP, which interpolates by computing weighted quantiles using class-dependent weights determined by some notion of class similarity.

## 3 Experiments

**Overview.** Code for reproducing our experiments is available at https://github.com/tiffanyding/long-tail-conformal. We consider two datasets for long-tailed classification: Pl@ntNet-300K (Garcin et al., 2021) and iNaturalist-2018 (Van Horn et al., 2018). Figure 2 shows the class distributions of the datasets we use. A key challenge of Pl@ntNet-300K and iNaturalist-2018 is that their test sets are also long-tailed, hindering reliable class-conditional evaluation for rare classes.[1] To address this, we create truncated versions (see Appendix C.1 for details) that preserve some of the long-tail structure while allowing for robust estimation of class-conditional metrics: Classes in the truncated datasets have 100 test examples each, but we assume the test distribution of interest is equivalent to the (long-tailed) train distribution, so we compute marginal metrics by computing a weighted average of the class-conditional metrics where the weight for class $y$ is the prevalence $\hat{p}(y)$ computed on train. When it comes to aggregated class-conditional metrics (such as macro-coverage), the full and truncated datasets produce similar results, so we defer the truncated results to Appendix D. However, when reporting unaggregated class-conditional metrics (as is the case in Section 3.3 below), it is necessary to use the truncated versions.

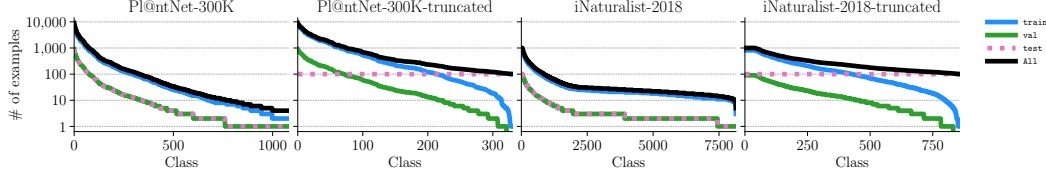

Figure 2: Class distributions (sorted by prevalence), plotted using a logarithmic scale, of the classical train, val, and test sets in the datasets we experiment on. We further randomly split 30% of val to use for model validation and use the remaining 70% as the calibration set $\mathcal{D}_{\text{cal}}$. We use the truncated versions to obtain good estimates of class-conditional metrics.

Unless otherwise stated, we use the $s_{\text{softmax}}$ score function described in Section 1.2. The base model is a ResNet-50 (He et al., 2016) trained using the standard cross-entropy loss (see Appendix D.3 for similar results using focal loss (Lin et al., 2017), a loss designed for class-imbalanced settings). More details about the experimental setup are available in Appendix C.

---

[1]69% of classes in Pl@ntNet-300K and 90% of classes in iNaturalist have fewer than 10 test examples.

**Metrics.** We evaluate each set-generating procedure $\mathcal{C} : \mathcal{X} \to 2^{\mathcal{Y}}$ on a test dataset $\{(X_i^{\mathcal{T}}, Y_i^{\mathcal{T}})\}_{i=1}^N$ that is separate from the data used for model training (and validation) and calibration. Let $\mathcal{J}_y \subseteq [N]$ be the set of indices of the test examples with label $y$, and define

$$\hat{c}_y := \frac{1}{|\mathcal{J}_y|} \sum_{i \in \mathcal{J}_y} \mathbb{1}\{Y_i^{\mathcal{T}} \in \mathcal{C}(X_i^{\mathcal{T}})\} \qquad (16)$$

as the empirical class-conditional coverage of class $y$.

We consider several ways of aggregating $\hat{c}_y$ across $y$ to obtain a scalar metric: *(i)* the fraction of classes with coverage below a threshold (50% in our experiments), *(ii)* the undercoverage gap, defined as the average undercoverage across classes (zero for classes with coverage at least $1 - \alpha$), and *(iii)* the average of $\hat{c}_y$'s, yielding the empirical macrocoverage. This is summarized in the first three rows of Table 2. Which of these metrics is most natural depends on the goal of the practitioner. Furthermore, we consider the standard prediction set metrics of marginal coverage and average set size, as detailed in the last two rows of Table 2.

Table 2: Our evaluation metrics. $\hat{c}_y$ is the empirical coverage for class $y$ on the test set of $N$ points, $1 - \alpha$ is the target coverage level, and $|\mathcal{Y}|$ is the number of classes.

| Metric name | Definition |
|---|---|
| FracBelow50% | $\frac{1}{|\mathcal{Y}|} \sum_{y \in \mathcal{Y}} \mathbb{1}\{\hat{c}_y \leq 0.5\}$ |
| UnderCovGap | $\frac{1}{|\mathcal{Y}|} \sum_{y \in \mathcal{Y}} \max(1 - \alpha - \hat{c}_y, 0)$ |
| MacroCov | $\frac{1}{|\mathcal{Y}|} \sum_{y \in \mathcal{Y}} \hat{c}_y$ |
| MarginalCov | $\frac{1}{N} \sum_{i=1}^N \mathbb{1}\{Y_i^{\mathcal{T}} \in \mathcal{C}(X_i^{\mathcal{T}})\}$ |
| Average set size | $\frac{1}{N} \sum_{i=1}^N |\mathcal{C}(X_i^{\mathcal{T}})|$ |

**Methods.** We run STANDARD with the PAS score function from Section 2.1. and INTERP-Q from Section 2.2 with weights $\tau \in \{0, 0.25, 0.5, 0.75, 0.9, 0.95, 0.975, 0.99, 0.999, 1\}$ on the CLASSWISE thresholds. These weights are chosen to effectively trace out the trade-off between set size and class-conditional coverage in our experiments. We compare against the following baseline conformal prediction methods: STANDARD, as described in (4); CLASSWISE, as described in (6); and CLUSTERED (Ding et al., 2023), a method that targets class-conditional coverage in the many-classes setting by grouping together classes with similar score distributions and computing a single score threshold for each cluster. Additional baselines with weaker performance, such as RC3P from Shi et al. (2024), are deferred to Appendix D.

## 3.1 Evaluating the size-coverage trade-off

Figure 3 visualizes the trade-off between set size and various notions of coverage (class-conditional, macro-, and marginal) achieved by each method. We describe some high-level takeaways:

*(i) When targeting set size and class-conditional or macro-coverage, it is more effective to optimize for this trade-off directly than trading off set size and marginal coverage.* Adjusting $\alpha$ in STANDARD CP is a plausible way to target class-conditional coverage but the results show that this does not optimally navigate the set size/class-conditional coverage trade-off, which is our goal. In comparison, our methods, which explicitly optimize for class-conditional or macro-coverage, consistently achieve better trade-offs than STANDARD CP.

*(ii) CLASSWISE should generally be avoided*, as comparable class-conditional and macro-coverage can be achieved with significantly smaller sets using our proposed methods.

*(iii) INTERP-Q produces reasonable set sizes even for large values of $\tau$.* Linearly interpolating between the STANDARD quantile and CLASSWISE quantile does not linearly interpolate the average set sizes of the two methods: at $\tau = 1$, INTERP-Q coincides with CLASSWISE and consequently has a very large average set size (780 for Pl@ntNet-300K and 7430 for iNaturalist-2018, $\alpha = 0.1$), but decreasing $\tau$ only slightly to $\tau = 0.99$ results in much more reasonable average set sizes of 7.6 on Pl@ntNet-300K and 55.8 on iNaturalist-2018. This nonlinear relationship is likely due to the fact that the $s_{\mathsf{softmax}}$ distribution of rare classes is highly skewed towards one because the classifier consistently assigns them predicted probabilities near zero.

*(iv) STANDARD with PAS is Pareto optimal*, in the sense that at any marginal coverage level, there is no method that simultaneously achieves better set size and class-conditional/macro- coverage. This suggests that STANDARD with PAS is a good starting place for practitioners due to its simplicity and strong performance on all metrics. However, INTERP-Q is also of practical value since its tunable

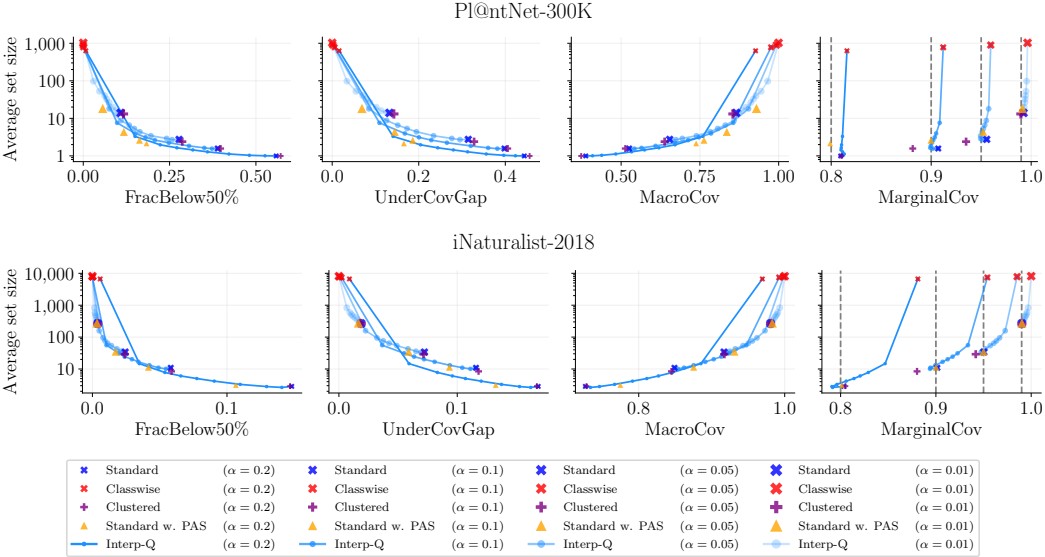

Figure 3: Average set size vs. FracBelow50%, UnderCovGap, MacroCov, and MarginalCov for various methods on the two datasets. For INTERP-Q, lines are used to trace out the trade-off curve achieved by running the method with different $\tau$ values for a fixed $\alpha$. For FracBelow50% and UnderCovGap, it is better to be closer to the bottom left. For MacroCov, the bottom right is better. For MarginalCov, we want to be to the right of the dotted line at $1 - \alpha$ for the $\alpha$ at which the method is run.

parameter allows practitioners to choose where they want to be on the trade-off curve between set size and class-conditional coverage while maintaining marginal coverage. Note that the two methods can also be combined, as presented in Appendix D.2, Figure 9.

**Pl@ntNet-300K case study.** Suppose we want sets with 90% marginal coverage on Pl@ntNet-300K. STANDARD has a small average set size of 1.57 but 421 of the 1081 plant species have coverage below 50%. Conversely, the CLASSWISE sets have zero classes with coverage below 50%, but an average set size of 780. Our methods provide a middle ground: STANDARD with PAS produces sets that have an average size only slightly bigger than STANDARD (2.57) but more than halves the number of classes with coverage below 50% to 180. INTERP-Q behaves similarly, with the added bonus that the trade-off between set size and class-conditional coverage can be tuned by adjusting $\tau$. We provide a table of the metric values plotted in Figure 3 in Appendix D.1 and an extended Pl@ntNet-300K case study in Appendix D.4.

### 3.2 TARGETING ENDANGERED SPECIES

Motivated by plant conservation, we use the weighted prevalence-adjusted softmax (WPAS) score to target coverage of at-risk species in Pl@ntNet-300K.[2] Let $\mathcal{Y}_{\text{at-risk}} \subseteq \mathcal{Y}$ be the set of at-risk species. We will weight the coverage of at-risk species $\lambda \geq 1$ times more than the coverage of other species, so

$$\omega(y) = \begin{cases} \frac{\lambda}{W} & \text{if } y \in \mathcal{Y}_{\text{at-risk}} \\ \frac{1}{W} & \text{otherwise,} \end{cases}$$

where $W = \lambda|\mathcal{Y}_{\text{at-risk}}| + |\mathcal{Y} \setminus \mathcal{Y}_{\text{at-risk}}|$ is a normalizing factor to ensure $\sum_{y \in \mathcal{Y}} \omega(y) = 1$.

The results are shown in Figure 4. We observe that STANDARD with WPAS improves the class-conditional coverage of at-risk classes relative to STANDARD with softmax or PAS. Comparing WPAS to PAS, we see that increasing $\lambda$, the amount we upweight at-risk classes, leads to larger improvements in the class-conditional coverage of at-risk classes, as expected. These increases are "paid for" in terms of a mild increase in average set size and have no discernible effect on the class-conditional coverage of not-at-risk classes, which is appealing from a practical perspective.

---

[2]We consider species with an IUCN status of "endangered", "vulnerable", "near threatened", "critically endangered", or "lower risk" as *at risk*. Of the 1,081 total species in Pl@ntNet-300K, 33 species qualify as at-risk.

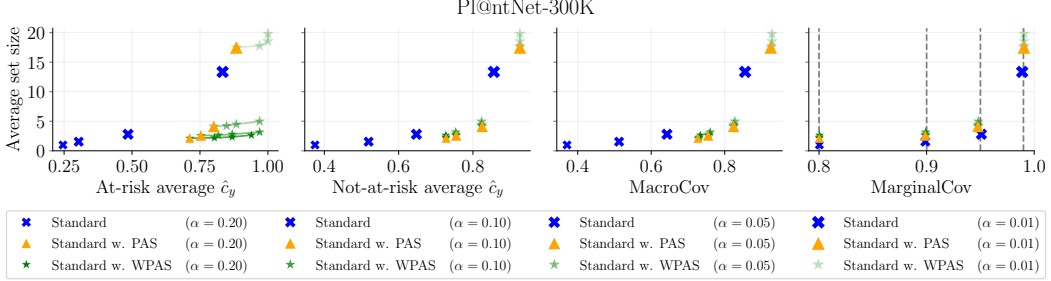

Figure 4: Results for running STANDARD on Pl@ntNet-300K with different conformal score functions: softmax, PAS, and WPAS with $\lambda \in \{1, 10, 10^2, 10^3\}$. Increasing $\lambda$ in WPAS improves the class-conditional coverage of at-risk classes, which is measured using $\hat{c}_y$, the empirical class-conditional coverage of class $y$. "At-risk average $\hat{c}_y$" is computed as $(1/|\mathcal{Y}_{\text{at-risk}}|) \sum_{y \in \mathcal{Y}_{\text{at-risk}}} \hat{c}_y$ and "not-at-risk average $\hat{c}_y$" is computed analogously. Note that here the y-axis is on a linear scale.

## 3.3 SIMULATED HUMAN DECISION-MAKER

We now examine how coverage and set size can jointly influence human decision accuracy. Human interpretations of prediction sets can vary (Zhang et al., 2024; Hullman et al., 2025), and we focus on two models of human decision-making that are impacted by coverage and set size: an *expert verifier* $H_{\text{expert}}$ and a *random guesser* $H_{\text{random}}$ (equivalent to the uncertainty suppressing decision maker in Hullman et al., 2025). Let $H(\mathcal{C}(X), Y) \in \mathcal{Y}$ be a human's chosen label given prediction set $\mathcal{C}(X)$ when the true label is $Y$. The probability $\mathbb{P}(H(\mathcal{C}(X), Y) = Y)$ that the human chooses the correct label after seeing prediction set $\mathcal{C}(X)$ is $\mathbb{1}\{Y \in \mathcal{C}(X)\}$ if $H = H_{\text{expert}}$ and $\mathbb{1}\{Y \in \mathcal{C}(X)\}/|\mathcal{C}(X)|$ if $H = H_{\text{random}}$. $H_{\text{expert}}$ is only affected by coverage and not set size, whereas random guessers are highly sensitive to set size, as they choose a label uniformly at random from the prediction set. We also consider mixture decision-makers, $H_{\text{mixture}}$, who act as $H_{\text{expert}}$ with probability $\gamma_{\text{exp.}}$ and $H_{\text{random}}$ with $1 - \gamma_{\text{exp.}}$ for $\gamma_{\text{exp.}} \in [0, 1]$. We are interested in how prediction sets affect the probability that a human correctly labels an instance of a given class, which we formalize as the *class-conditional decision accuracy* for class $y$ under procedure $\mathcal{C}$, defined as $\mathbb{P}(H(\mathcal{C}(X), Y) = Y \mid Y = y)$. Note that, due to the definition of $H_{\text{expert}}$, the class-conditional decision accuracy of $H_{\text{expert}}$ is equivalent to the class-conditional coverage.

Figure 5 shows the class-conditional decision accuracies for various types of decision makers under STANDARD with PAS as compared to baseline methods (the plots for INTERP-Q are similar; see Appendix D.6). We report results on the truncated versions of Pl@ntNet-300K and iNaturalist-2018 that only include classes with enough test examples to reliably estimate class-conditional decision accuracy (see Appendix C.1 for more details). We observe that STANDARD and STANDARD with PAS achieve strong performance regardless of $\gamma_{\text{exp.}}$, whereas CLASSWISE only does well when $\gamma_{\text{exp.}}$ is high. However, compared to STANDARD, STANDARD with PAS has an additional benefit that its performance is more balanced across classes. The worst decision accuracies for STANDARD with PAS are higher than those of STANDARD, which is paid for by only a slight decrease in the best decision accuracies.

## 4 DISCUSSION

In this paper, we consider the problem of producing good prediction sets in long-tailed settings, which heretofore has been an understudied problem despite its relevance in many applications, such as species or disease classification. We develop two approaches to better navigate the trade-off between set size and class-conditional coverage in long-tailed settings. First, we introduce prevalence-adjusted softmax (PAS) and its weighted variant (WPAS), score functions inspired by oracle solutions that optimally balance expected set size and (weighted) macro-coverage. Second, we propose INTERP-Q, a simple procedure that interpolates between STANDARD and CLASSWISE CP to provide a tunable compromise between small sets and strong class-conditional coverage.

*Limitations and future work.* To use INTERP-Q, we must choose a value of $\tau$. In practice, if we select a desired average set size and choose the parameter value that achieves this size on the calibration set, this will produce reasonable results (despite the exchangeability violation). Another

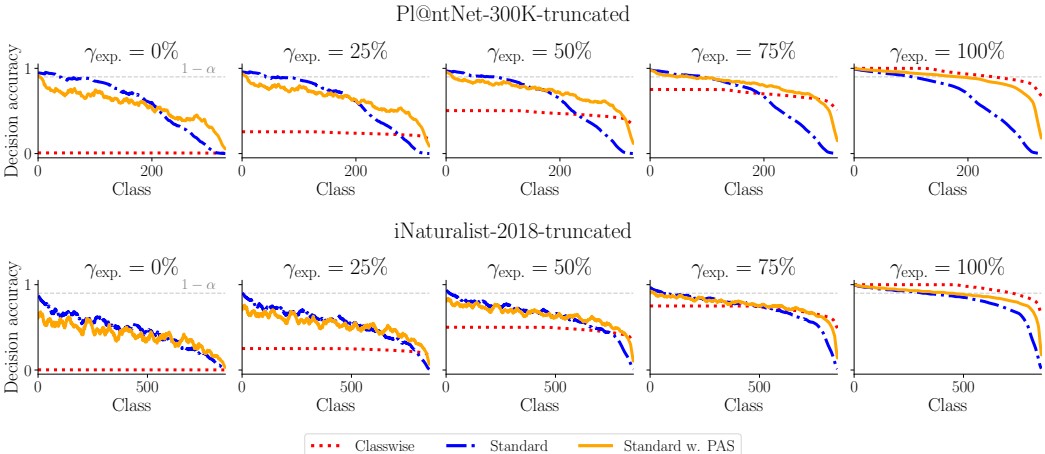

Figure 5: Class-conditional decision accuracies for a range of decision makers when presented with sets from STANDARD, CLASSWISE or STANDARD with PAS at $\alpha = 0.1$ . Classes are ordered by decreasing decision accuracy of $H_{\text{expert}}$ under each method. Note that the decision accuracy when $\gamma_{\text{exp.}} = 100\%$ is equivalent to the class-conditional coverage.

promising future research direction that we touched upon is the interaction between set sizes and coverage in determining the utility of prediction sets. In particular, an important aspect of decision-making not addressed by the decision accuracy discussed in Section 3.3 is the concept of "effort": even if a human can identify the correct label within a set, larger sets require more effort to search through.

*Broader impacts.* The methods we introduce have the potential to benefit society in the mid- to long-term by improving uncertainty quantification on citizen science platforms like Pl@ntNet. Our methods increase the probability that prediction sets for rare or endangered species contain the true label while keeping the average set size under control. This has important implications for biodiversity monitoring and has the potential to produce a virtuous cycle: As more images of rare species are identified, they can be used to retrain the classifier to better identify such species, which in turn improves the probability that citizen scientists will correctly classify future images of these species.

## ACKNOWLEDGMENTS

We thank Pierre Bonnet, Alexis Joly, Anastasios Angelopolous, and Ryan Tibshirani for insightful conversations and Margaux Zaffran for making this project possible. This work was funded by the French National Research Agency (ANR) through the grant Chaire IA CaMeLOt (ANR-20-CHIA-0001-01). TD additionally acknowledges support from the National Science Foundation Graduate Research Fellowship Program under grant no. 2146752.

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

## A FUZZY CLASSWISE CONFORMAL PREDICTION

The INTERP-Q method introduced in Section 2.2 shrinks $\hat{q}_y$ in a naive way that disregards relationships between classes; in this section, we propose a more sophisticated approach called *fuzzy classwise CP* (FUZZY for short) that assigns weights to the score of each calibration example based on the "similarity" between their class and class $y$ and then takes the quantile of this weighted distribution. Notably, this approach comes with a $1 - \alpha$ marginal coverage guarantee (rather than the $1 - 2\alpha$ guarantee of INTERP-Q). Empirically, we find that it does not do better than the simpler INTERP-Q procedure, but we still believe it is useful to present the method, as it may be preferable in settings where a marginal coverage guarantee is desired. Proofs are deferred to Appendix B.

This method relates to work on weighted conformal prediction, which generalizes standard conformal prediction by computing a *weighted* quantile of calibration scores (Tibshirani et al., 2019; Barber et al., 2023; Barber & Tibshirani, 2025). Methodologically closest to our work is Podkopaev & Ramdas (2021), which uses label-dependent weighting to ensure marginal coverage under label shift.

### A.1 PRELIMINARIES

Recall the STANDARD CP and CLASSWISE CP procedures introduced in Section 1.2. We now consider a generalization of these two methods. Let $w : \mathcal{Y} \times \mathcal{Y} \to \mathbb{R}_{\geq 0}$ be a weighting function where $w(y', y)$ is the weight assigned to a point with label $y'$ when computing the weighted quantile threshold for class $y$. The *w-label-weighted conformal prediction set* (see, e.g., Podkopaev & Ramdas, 2021) is $\mathcal{C}_{\mathrm{LW}}(X; w) := \mathcal{C}(X; \hat{\mathbf{q}}_w)$ where the $y$-th entry of $\hat{\mathbf{q}}_w$ is

$$\hat{q}_y^w = \mathrm{Quantile}_{1-\alpha}\Big( \sum_{i=1}^{n} \frac{w(Y_i, y)}{W_y}\delta_{s(X_i, Y_i)} + \frac{w(y, y)}{W_y}\delta_\infty \Big) \tag{17}$$

and $W_y = \sum_{i=1}^{n} w(Y_i, y) + w(y, y)$ is a normalizing factor to ensure the weights sum to one.

STANDARD corresponds to label-weighted conformal with equal weights for all classes regardless of $y$ — that is, $w(y', y) \propto 1$. CLASSWISE corresponds to label-weighted conformal where nonzero weights are assigned only to other calibration points of class $y$, i.e. $w(y', y) \propto \mathbb{1}\{y' = y\}$.

### A.2 THE FUZZY METHOD

We now present the *fuzzy classwise conformal prediction* (FUZZY) procedure. We begin by presenting a simpler method, Raw FUZZY, that FUZZY builds upon.

**Raw FUZZY.** The large sets of CLASSWISE result from not having enough data from each class when computing the score quantiles. A natural solution to this problem is to borrow data from other classes, where more data is borrowed from classes that are more similar.

We define the *Raw FUZZY prediction set* to be the label-weighted conformal prediction set obtained using $w_{\mathrm{FUZZY}} : \mathcal{Y} \times \mathcal{Y} \to \mathbb{R}_{\geq 0}$ as the weighting function in (17), i.e., $\mathcal{C}_{\mathrm{RAWFUZZY}}(X) := \mathcal{C}_{\mathrm{LW}}(X; w_{\mathrm{FUZZY}})$.

To construct this weighting function, we use a class mapping $\Pi : \mathcal{Y} \to \Lambda \subseteq \mathbb{R}^d$ for some small $d > 0$ (e.g., $\Lambda = (0, 1)$) and kernel functions $h_y^\sigma : \Lambda \times \Lambda \to \mathbb{R}_{\geq 0}$ parameterized by a bandwidth $\sigma > 0$.

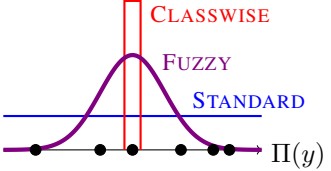

Figure 6: The weighting function $w$ of each CP method, when viewed as label-weighted conformal prediction.

A good $\Pi$ should map classes with similar score distributions close together. The kernel $h_y^\sigma$ takes in two mapped classes and outputs a non-negative scalar that is decreasing with the distance between the two inputs, and it is allowed to depend on $y$. For example, the kernel bandwidth could be rescaled to decrease with the number of examples we have from class $y$ so that classes with more examples "borrow less" from other classes.

We then define the weighting function as

$$w_{\mathrm{FUZZY}}(y', y) = h_y^\sigma\big(\Pi(y'), \Pi(y)\big). \tag{18}$$

Figure 6 visualizes how Raw FUZZY can be viewed as interpolating between STANDARD and CLASSWISE CP by setting weights that are in between the two extremes.

The following proposition tells us that for most reasonable kernels, such as the Gaussian kernel $h^\sigma(u, v) \propto \exp\left(-(v-u)^2/(2\sigma^2)\right)$, Raw FUZZY can recover both STANDARD and CLASSWISE by setting the bandwidth appropriately.

**Proposition 4** (Informal). *Assume $\Pi$ maps each class to a unique point. If, for all $u, v \in \Lambda$ with $u \neq v$, the kernel $h$ satisfies $h_\sigma(u, v) \to 0$ as $\sigma \to 0$ and $h_\sigma(u, v) \to h_\sigma(u, u)$ as $\sigma \to \infty$, then for sufficiently small $\sigma$ we have $\mathcal{C}_{\text{RAWFUZZY}}(X) \equiv \mathcal{C}_{\text{CLASSWISE}}(X)$ and for sufficiently large $\sigma$ we have $C_{\text{RAWFUZZY}}(X) \equiv \mathcal{C}_{\text{STANDARD}}(X)$.*

**FUZZY = Raw FUZZY + reconformalization for marginal coverage.** To obtain the FUZZY procedure, we add a reconformalization wrapper around Raw FUZZY to ensure marginal coverage. In order to do this reconformalization, we use a holdout dataset. In practice, this holdout dataset can be created by setting aside a relatively small part of the calibration dataset (here, 5000 examples) since we only use it to estimate a single parameter. The intuition behind our procedure is as follows. When reconformalizing, we hope to equally affect all class-conditional coverages. One way to do this is to find the $\tilde{\alpha} \geq 0$ such that running Raw FUZZY at level $1 - \tilde{\alpha}$ achieves the desired $1 - \alpha$ coverage. This can be formulated as an example of STANDARD CP with a special score function $\tilde{s}(x, y) = \hat{F}_y\big(s(x, y)\big)$, where

$$\hat{F}_y(t) = \sum_{i=1}^n w_i^y \mathbb{1}\{s(X_i, Y_i) < t\} \quad \text{and} \quad w_i^y = \frac{w_{\text{FUZZY}}(Y_i, y)}{\sum_{i=1}^n w_{\text{FUZZY}}(Y_i, y) + w_{\text{FUZZY}}(y, y)}. \tag{19}$$

Then, we have the equivalence $y \in \mathcal{C}_{\text{RAWFUZZY}}(x) \iff \tilde{s}(x, y) < 1 - \alpha$. The score $\tilde{s}$ can be interpreted as (one minus) a weighted conformal p-value (Vovk et al., 2005; Barber & Tibshirani, 2025), which we re-calibrate. To achieve the desired marginal coverage, the method FUZZY does STANDARD CP with the new score function $\tilde{s}$ and held-out data as a calibration set.

**Proposition 5.** *Let $\mathcal{D}_{\text{cal}} = \{(X_i, Y_i)\}_{i=1}^n$ be a calibration set and $\mathcal{D}_{\text{hold}} = \{(X_i^{\mathcal{H}}, Y_i^{\mathcal{H}})\}_{i=1}^m$ be held-out examples. Define $\tilde{\alpha}$ to be the STANDARD CP threshold obtained by applying $\tilde{s}$ on the holdout set:*

$$1 - \tilde{\alpha} = \text{Quantile}_{1-\alpha}\Big(\frac{1}{m+1}\sum_{i=1}^m \delta_{\tilde{s}(X_i^{\mathcal{H}}, Y_i^{\mathcal{H}})} + \frac{1}{m+1}\delta_\infty\Big). \tag{20}$$

*Then, if the scores evaluated on the held-out dataset $\tilde{s}(X_i^{\mathcal{H}}, Y_i^{\mathcal{H}})$ are exchangeable with the score of the test point, the set $\mathcal{C}_{\text{FUZZY}}(x) = \{y : \tilde{s}(x, y) \leq 1 - \tilde{\alpha}\}$ will achieve $1 - \alpha$ marginal coverage.*

The score function $\tilde{s}$ depends on the calibration set $\mathcal{D}_{\text{cal}}$, but the assumption of exchangeability is easily satisfied, e.g., if the points of the calibration set and the held-out set and the test point are exchangeable. A careful reader may also notice that $\mathcal{C}_{\text{FUZZY}}$ is not exactly equivalent to apply Raw FUZZY at level $\tilde{\alpha}$ as a strict inequality has been replaced by a non-strict one to get the coverage guarantee. As a result, we implement FUZZY as Raw FUZZY at level $\tilde{\alpha} - \varepsilon$ for a small perturbation $\varepsilon > 0$ (specifically, $\varepsilon < \min_{i,y} \omega_i^y$).

Note that, if desired, we could use full conformal techniques (Vovk et al., 2005) instead of data splitting (i.e., let $\mathcal{D}_{\text{cal}} = \mathcal{D}_{\text{hold}}$), but this incurs higher computational costs that are undesirable for practical applications (see Proposition 8 for details).

**Choosing a mapping $\Pi$.** To instantiate FUZZY, we must define a mapping $\Pi$ from $\mathcal{Y}$ to a space in which we can compute distances between classes. What makes a good mapping? Intuitively, we want classes with similar score distributions to be mapped close together. This is similar to the motivation behind the clustering procedure from Ding et al. (2023). However, in long-tailed settings, many classes do not have sufficient calibration examples for us to estimate their score distributions. A "zero-shot" way of attempting to group together classes with similar score distributions is to group together classes based on their prevalence, with the intuition that the underlying classifier is likely to assign small softmax scores to infrequently seen classes and large scores to more frequently seen classes. Specifically, we map each class $y$ to its number of `train` examples $n_y^{\text{train}}$, normalize, and then add a small amount of random noise to ensure that the uniqueness condition of Proposition 4 is satisfied with probability one: $\Pi_{\text{prevalence}}(y) = cn_y^{\text{train}} + \varepsilon_y$ where $\varepsilon_y \sim \text{Unif}([-0.01, 0.01])$ independently for each for $y \in \mathcal{Y}$. We normalize using $c = 1/(\max_{y' \in \mathcal{Y}} n_{y'}^{\text{train}})$ to ensure that $\Pi_{\text{prevalence}}(y) \in [0, 1]$ so that the same bandwidth $\sigma$ has similar effects on different datasets. This is just one possible choice for $\Pi$ and is what we use in our main experiments; other options are explored in Appendix D.2.

## B   THEORETICAL GUARANTEES

### B.1   OPTIMAL PREDICTION SETS FOR (WEIGHTED) MACRO-COVERAGE

In this section, we state and prove a more formal version of Proposition 2, of which Proposition 1 is a special case with uniform weights.

**Proposition 6** (Formal version of Proposition 2). *Let $\omega : \mathcal{Y} \to [0,1]$ be a non-negative weighting function summing to one. For $t \in \mathbb{R}$, define*

$$\tilde{\mathcal{C}}_t(x) = \{y \in \mathcal{Y} : \tilde{s}(x,y) \geq t\} \quad where \quad \tilde{s}(x,y) = \frac{\omega(y)}{p(y)} \cdot p(y|x) \tag{21}$$

*and $p(y|x)$ denotes the conditional density of $Y$ given $X = x$ and $p(y)$ is the marginal density of $Y$.*

*(a) Let $\alpha \in [0,1]$. If there exists $t_\alpha$ such that $\mathrm{MacroCov}_\omega(\tilde{\mathcal{C}}_{t_\alpha}) = 1 - \alpha$, then $\tilde{\mathcal{C}}_{t_\alpha}$ is the optimal solution to*

$$\min_{\mathcal{C}:\mathcal{X} \mapsto 2^{\mathcal{Y}}} \mathbb{E}[|\mathcal{C}(X)|] \quad subject \ to \ \sum_{y \in \mathcal{Y}} \omega(y) \mathbb{P}\left(y \in \mathcal{C}(X) \mid Y = y\right) \geq 1 - \alpha. \tag{22}$$

*(b) Let $\kappa \geq 0$. If there exists $t_\kappa$ such that $\mathbb{E}[|\tilde{\mathcal{C}}_{t_\kappa}|] = \kappa$, then $\tilde{\mathcal{C}}_{t_\kappa}$ is the optimal solution to*

$$\max_{\mathcal{C}:\mathcal{X} \mapsto 2^{\mathcal{Y}}} \sum_{y \in \mathcal{Y}} \omega(y) \mathbb{P}\left(y \in \mathcal{C}(X) \mid Y = y\right) \quad subject \ to \ \mathbb{E}[|\mathcal{C}(X)|] \leq \kappa. \tag{23}$$

**Remark 1.** If thresholds $t_\alpha$ or $t_\kappa$ achieving exact equality do not exist, the optimal set remains of the form (21) but must be combined with randomization to achieve the optimal solution (that has weighted macro-coverage of exactly $1 - \alpha$ or expected set size of exactly $\kappa$). See, for instance, Shao (2008, Theorem 6.1) for a formal statement of Lemma B.1 below for this case.

*Proof.* This result is a consequence of the Neyman-Pearson Lemma, which we state below using the formulation of Sadinle et al. (2019) (see also Casella & Berger, 2001, Theorem 8.3.12). We provide a proof of this lemma in Appendix B.2.

**Lemma B.1** (Neyman & Pearson, 1933). *Let $\mu$ be a measure on $\mathcal{X} \times \mathcal{Y}$ and let $f, g : \mathcal{X} \times \mathcal{Y} \to \mathbb{R}_{\geq 0}$ be two non-negative functions. For $\nu \geq 0$, consider the problem*

$$\min_{\mathcal{C}:\mathcal{X} \to 2^{\mathcal{Y}}} \int_{\mathcal{X} \times \mathcal{Y}} \mathbb{1}\{y \in \mathcal{C}(x)\} g(x,y) d\mu(x,y) \tag{24}$$

$$subject \ to \ \int_{\mathcal{X} \times \mathcal{Y}} \mathbb{1}\{y \in \mathcal{C}(x)\} f(x,y) d\mu(x,y) \geq \nu.$$

*If there exists $t_\nu$ such that*

$$\int_{\mathcal{X} \times \mathcal{Y}} \mathbb{1}\{f(x,y) \geq t_\nu \cdot g(x,y)\} f(x,y) d\mu(x,y) = \nu, \tag{25}$$

*then*

$$\mathcal{C}^*_\nu(x) = \{y \in \mathcal{Y} : f(x,y) \geq t_\nu \cdot g(x,y)\} \tag{26}$$

*is the optimal solution to (24): Any other minimizer $\mathcal{C}$ of (24) is equal to $\mathcal{C}^*_\nu$ $\mu_f$ and $\mu_g$-almost everywhere, i.e., for $h \in \{f, g\}$, $\mu_h\big(\{(x,y) : y \in \mathcal{C}^*_\nu(x)\} \Delta \{(x,y) : y \in \mathcal{C}(x)\}\big) = 0$ where $\Delta$ denotes the symmetric distance between the sets and $\mu_h$ is defined as $\mu_h(A) = \int_A h d\mu$.*

**Remark 2.** An analogous statement of Lemma B.1 holds where we replace the "$\geq$" with "$>$" in (25) and (26).

PROOF OF PROPOSITION 6(a). We assume that $X$ has a density $p$ with respect to the Lebesgue measure $\lambda$, but the proof can be adapted for any measure. Define the functions $f(x,y) := \omega(y)p(x|y)$ and $g(x,y) = p(x)$ and the measure $\mu = \lambda \times \left(\sum_{y \in \mathcal{Y}} \delta_y\right)$, which is the product measure between

the Lebesgue measure (denoted by $\lambda$) and the counting measure on $\mathcal{Y}$. Let $\nu = 1 - \alpha$. Observe that plugging these values of $f$, $g$, $\mu$, and $\nu$ into (24) yields (22), because

$$\int_{\mathcal{X} \times \mathcal{Y}} \mathbb{1}\{y \in \mathcal{C}(x)\} f(x,y) d\mu(x,y) = \sum_{y \in \mathcal{Y}} \omega(y) \int_{\mathcal{X}} \mathbb{1}\{y \in \mathcal{C}(x)\} p(x|y) dx$$

$$= \sum_{y \in \mathcal{Y}} \omega(y) \mathbb{P}\left(y \in \mathcal{C}(X) \mid Y = y\right),$$

and

$$\int_{\mathcal{X} \times \mathcal{Y}} \mathbb{1}\{y \in \mathcal{C}(x)\} g(x,y) d\mu(x,y) = \sum_{y \in \mathcal{Y}} \int_{\mathcal{X}} \mathbb{1}\{y \in \mathcal{C}(x)\} p(x) dx$$

$$= \int_{\mathcal{X}} \sum_{y \in \mathcal{Y}} \mathbb{1}\{y \in \mathcal{C}(x)\} p(x) dx$$

$$= \int_{\mathcal{X}} |\mathcal{C}(X)| p(x) dx$$

$$= \mathbb{E}\left(|\mathcal{C}(X)|\right).$$

By Lemma B.1, it follows that the optimal solution to (22) is given by (26), which for our choice of $f$ and $g$ can be rewritten as $\mathcal{C}^*_{1-\alpha}(x) = \{y \in \mathcal{Y} : \frac{\omega(y)p(y|x)}{p(y)} \geq t_{1-\alpha}\}$ by observing that

$$\frac{f(x,y)}{g(x,y)} = \frac{\omega(y)p(x|y)}{p(x)} = \frac{\omega(y)p(x,y)}{p(x)p(y)} = \frac{\omega(y)p(y|x)}{p(y)}.$$

PROOF OF PROPOSITION 6(b). Let us now derive the optimal solution to the dual problem. To do so, we must rewrite the problem in a form where Lemma B.1 can be applied. First, observe that

$$\sum_{y \in \mathcal{Y}} \omega(y) \mathbb{P}\left(y \in \mathcal{C}(X)|Y = y\right) = \sum_{y \in \mathcal{Y}} \omega(y)\left(1 - \mathbb{P}\left(y \notin \mathcal{C}(X) \mid Y = y\right)\right)$$

$$= 1 - \sum_{y \in \mathcal{Y}} \omega(y) \mathbb{P}\left(y \in \mathcal{C}^c(X) \mid Y = y\right),$$

where $\mathcal{C}^c(X) := \mathcal{Y} \setminus \mathcal{C}(X)$ denotes the complement of $\mathcal{C}(X)$. Similarly, observe that expected set size can be written in terms of the complement as $\mathbb{E}(|\mathcal{C}(X)|) = |\mathcal{Y}| - \mathbb{E}(|\mathcal{C}^c(X)|)$. Thus, we can obtain the optimal solution to (23) by taking the complement of the optimal solution to

$$\min_{\bar{\mathcal{C}}:\mathcal{X} \mapsto 2^{\mathcal{Y}}} \sum_{y \in \mathcal{Y}} \omega(y) \mathbb{P}\left(y \in \tilde{\mathcal{C}}(X) \mid Y = y\right) \quad \text{subject to} \quad \mathbb{E}(|\bar{\mathcal{C}}(X)|) \geq |\mathcal{Y}| - \kappa. \qquad (27)$$

Applying Lemma B.1, combined with Remark 2, with $f(x,y) = p(x)$, $g(x,y) = \omega(y)p(x|y)$, the same measure $\mu$ as in the proof of Proposition 6(a), and $\nu = |\mathcal{Y}| - \kappa$ tells us that if there exists $\bar{t}_\kappa$ such that

$$\int_{\mathcal{X} \times \mathcal{Y}} \mathbb{1}\{f(x,y) > \bar{t}_\kappa \cdot g(x,y)\} f(x,y) d\mu(x,y) = |\mathcal{Y}| - \kappa, \qquad (28)$$

then the optimal solution to (27) is

$$\bar{\mathcal{C}}^*_\kappa(x) = \left\{y \in \mathcal{Y} : \frac{f(x,y)}{g(x,y)} > \bar{t}_\kappa\right\} = \left\{y \in \mathcal{Y} : \frac{p(y)}{\omega(y)p(y|x)} > \bar{t}_\kappa\right\} = \left\{y \in \mathcal{Y} : \frac{\omega(y)p(y|x)}{p(y)} < \bar{t}_\kappa^{-1}\right\}.$$

Thus, the optimal solution to our original problem (23) is

$$\mathcal{C}^*_\kappa(x) := (\bar{\mathcal{C}}^*_\kappa)^c(x) = \left\{y \in \mathcal{Y} : \frac{\omega(y)p(y|x)}{p(y)} \geq \bar{t}_\kappa^{-1}\right\}.$$

$\square$

### B.2    PROOF OF LEMMA B.1

For completeness we give a proof of the Neyman-Pearson Lemma, as stated in Lemma B.1.

*Proof.* We will show that $\mathcal{C}^*(x) = \{y : f(x,y) \geq t_\nu g(x,y)\}$ is the optimal solution to (24). We first demonstrate that it is an optimal solution and then that it is unique.

*Optimality.* By definition of $t_\nu$, we have

$$\int_{\mathcal{X}\times\mathcal{Y}} \mathbb{1}\{y \in \mathcal{C}^*(x)\} f(x,y) d\mu(x,y) = \nu, \tag{29}$$

which is trivially greater than or equal to $\nu$, so $\mathcal{C}^*(x)$ is indeed a feasible solution. To show that $\mathcal{C}^*(x)$ is optimal, we must argue that it achieves a smaller objective value than any other feasible solution. Let $\mathcal{C} : \mathcal{X} \to 2^{\mathcal{Y}}$ be any other set-generating procedure that satisfies the constraint in (24). We want to show that

$$\int_{\mathcal{X}\times\mathcal{Y}} \mathbb{1}\{y \in \mathcal{C}^*(x)\} g(x,y) d\mu(x,y) \leq \int_{\mathcal{X}\times\mathcal{Y}} \mathbb{1}\{y \in \mathcal{C}(x)\} g(x,y) d\mu(x,y) \,.$$

We prove this by showing their difference is negative:

$$\int_{\mathcal{X}\times\mathcal{Y}} \mathbb{1}\{y \in \mathcal{C}^*(x)\} g(x,y) d\mu(x,y) - \int_{\mathcal{X}\times\mathcal{Y}} \mathbb{1}\{y \in \mathcal{C}(x)\} g(x,y) d\mu(x,y)$$

$$= \int_{\mathcal{X}\times\mathcal{Y}} \mathbb{1}\{y \in \mathcal{C}^*(x)\backslash\mathcal{C}(x)\} g(x,y) d\mu(x,y) - \int_{\mathcal{X}\times\mathcal{Y}} \mathbb{1}\{y \in \mathcal{C}(x)\backslash\mathcal{C}^*(x)\} g(x,y) d\mu(x,y)$$

$$\leq \frac{1}{t_\nu} \int_{\mathcal{X}\times\mathcal{Y}} \mathbb{1}\{y \in \mathcal{C}^*(x)\backslash\mathcal{C}(x)\} f(x,y) d\mu(x,y) - \frac{1}{t_\nu} \int_{\mathcal{X}\times\mathcal{Y}} \mathbb{1}\{y \in \mathcal{C}(x)\backslash\mathcal{C}^*(x)\} f(x,y) d\mu(x,y)$$

$$= \frac{1}{t_\nu} \left[ \int_{\mathcal{X}\times\mathcal{Y}} \mathbb{1}\{y \in \mathcal{C}^*(x)\} f(x,y) d\mu(x,y) - \int_{\mathcal{X}\times\mathcal{Y}} \mathbb{1}\{y \in \mathcal{C}(x)\} f(x,y) d\mu(x,y) \right]$$

$$\leq \frac{1}{t_\nu} (\nu - \nu) \leq 0 \,.$$

The first inequality follows from $y \in \mathcal{C}^*(x) \iff g(x,y) \leq t_\nu^{-1} f(x,y)$. The second inequality comes from applying the equality stated in (29) to the first integral and then using the definition of $\mathcal{C}$ as satisfying the constraint of (24) to lower bound the second integral.

*Uniqueness.* Let $\mathcal{C} : \mathcal{X} \to 2^{\mathcal{Y}}$ be another optimal set-generating procedure, so it achieves the same objective value as $\mathcal{C}^*$,

$$\int_{\mathcal{X}\times\mathcal{Y}} \mathbb{1}\{y \in \mathcal{C}(x)\} g(x,y) d\mu(x,y) = \int_{\mathcal{X}\times\mathcal{Y}} \mathbb{1}\{y \in \mathcal{C}^*(x)\} g(x,y) d\mu(x,y) \,, \tag{30}$$

and it is a feasible solution,

$$\int_{\mathcal{X}\times\mathcal{Y}} \mathbb{1}\{y \in \mathcal{C}(x)\} f(x,y) d\mu(x,y) \geq \nu \,.$$

Let us first note the following non-negativity relationship:

$$\left(\mathbb{1}\{y \in \mathcal{C}^*(x)\} - \mathbb{1}\{y \in \mathcal{C}(x)\}\right)\left(f(x,y) - t_\nu g(x,y)\right) \geq 0, \quad \forall (x,y) \in \mathcal{X} \times \mathcal{Y}. \tag{31}$$

Integrating (31) over $\mathcal{X} \times \mathcal{Y}$, then applying (30), we get

$$\int_{\mathcal{X}\times\mathcal{Y}} \mathbb{1}\{y \in \mathcal{C}^*(x)\} f(x,y) d\mu(x,y) - \int_{\mathcal{X}\times\mathcal{Y}} \mathbb{1}\{y \in \mathcal{C}(x)\} f(x,y) d\mu(x,y) \geq 0 \,.$$

Combining this with the definition of $\mathcal{C}^*$ and the feasibility of $\mathcal{C}$, we must have $\int_{\mathcal{X}\times\mathcal{Y}} \mathbb{1}\{y \in \mathcal{C}(x)\} f(x,y) d\mu(x,y) = \nu$. Thus, (31) integrates to zero. Since we also know that (31) is non-negative, it must be true that, $\mu$-almost everywhere,

$$\left(\mathbb{1}\{y \in \mathcal{C}^*(x)\} - \mathbb{1}\{y \in \mathcal{C}(x)\}\right)\left(f(x,y) - t_\nu g(x,y)\right) = 0.$$

For $(x, y) \in \mathcal{X} \times \mathcal{Y}$ such that $f(x, y) \neq t_\nu g(x, y)$, then $\mathbb{1}\{y \in \mathcal{C}^*(x)\} = \mathbb{1}\{y \in \mathcal{C}(x)\}$, which implies, using the definition of $\mathcal{C}^*(x) = \{y : f(x, y) \geq t_\nu g(x, y)\}$, that $\mu$-almost everywhere, $\mathcal{C}(x) \subseteq \mathcal{C}^*(x)$ and $\mathcal{C}^*(x) \subseteq \mathcal{C}(x) \cup \{y : f(x, y) = t_\nu g(x, y)\}$. It remains to show that the sets are equal almost everywhere by showing that the set

$$D := \left\{(x, y) : y \notin \mathcal{C}(x) \text{ and } y \in \mathcal{C}^*(x)\right\} = \left\{(x, y) : y \notin \mathcal{C}(x) \text{ and } f(x, y) = t_\nu g(x, y)\right\}$$

is of measure $0$ under $\mu_f$. By definition of $\mathcal{C}^*$,

$$\begin{aligned}
\nu &= \int_{\mathcal{X} \times \mathcal{Y}} \mathbb{1}\{y \in \mathcal{C}^*(x)\} f(x, y) d\mu(x, y) \\
&= \int_{\mathcal{X} \times \mathcal{Y}} \mathbb{1}\{y \in \mathcal{C}(x)\} f(x, y) d\mu(x, y) + \int_{\mathcal{X} \times \mathcal{Y}} \mathbb{1}\{(x, y) \in D\} f(x, y) d\mu(x, y) \\
&= \nu + \int_D f d\mu \,.
\end{aligned}$$

Thus we must have $\mu_f(D) = 0$. Since $t_\nu g = f$ on $D$, we also get that $\mu_g(D) = 0$.

The version of the Neyman-Pearson Lemma described in Remark 2 can be proven in the same way. □

### B.3 PROOF OF PROPOSITION 3

We present Proposition 3 in two parts: the lower bound on coverage and the extreme case achieving a coverage close to this lower bound.

**Proposition 3** (restated, first part). *Using calibration data $\{(X_i, Y_i)\}_{i=1}^n$, let $\hat{q}$ be the STANDARD quantile threshold (4) computed for $\alpha \in [0, 1]$ and, for $y \in \mathcal{Y}$, let $\hat{q}_y^{\text{CW}}$ be the CLASSWISE quantile threshold (6) computed for the same $\alpha$. If the test point $(X_{n+1}, Y_{n+1})$ is exchangeable with the calibration data $\{(X_i, Y_i)\}_{i=1}^n$, then the INTERP-Q prediction sets satisfy*

$$\mathbb{P}(Y_{n+1} \in C_{\text{INTERP-Q}}(X_{n+1})) \geq 1 - 2\alpha. \tag{32}$$

*Proof.* Observe that for the test score $S_{n+1} = s(X_{n+1}, Y_{n+1})$, we have the following inclusion of events:

$$\left\{S_{n+1} \leq \tau \hat{q}_{Y_{n+1}}^{\text{CW}} + (1 - \tau)\hat{q}\right\} \supset \left\{S_{n+1} \leq \min(\hat{q}_{Y_{n+1}}^{\text{CW}}, \hat{q})\right\}.$$

Thus,

$$\begin{aligned}
\mathbb{P}\left(S_{n+1} \leq \tau \hat{q}_{Y_{n+1}}^{\text{CW}} + (1 - \tau)\hat{q}\right) &\geq \mathbb{P}\left(S_{n+1} \leq \min(\hat{q}_{Y_{n+1}}^{\text{CW}}, \hat{q})\right) \\
&= 1 - \mathbb{P}\left(S_{n+1} > \hat{q}_{Y_{n+1}}^{\text{CW}} \text{ or } S_{n+1} > \hat{q}\right) \\
&\geq 1 - \mathbb{P}\left(S_{n+1} > \hat{q}_{Y_{n+1}}^{\text{CW}}\right) - \mathbb{P}\left(S_{n+1} > \hat{q}\right),
\end{aligned}$$

using a union bound for the last inequality. By the coverage guarantees for STANDARD and CLASSWISE, which follow from classical conformal prediction arguments using exchangeability (see, e.g., Vovk et al., 2005), we have $\mathbb{P}\left(S_{n+1} > \hat{q}_{Y_{n+1}}^{\text{CW}}\right) \leq \alpha$ and $\mathbb{P}(S_{n+1} > \hat{q}) \leq \alpha$, which concludes the proof. □

The following result provides an example of a score distribution for which a coverage of $(1 - \alpha)^2$ is attained. For simplicity, the example is constructed as if the distribution and the quantile are exactly known, i.e., in the case of an infinite calibration set ($n = \infty$).

**Proposition 3** (restated, second part). *There exists a joint distribution $\mathbb{P}$ of tuple (score, label) over $[0, 1] \times \mathcal{Y}$ with $|\mathcal{Y}| \geq 2$, such that for any $\tau \in (0, 1)$, we have*

$$\mathbb{P}\left(S \leq \tau q_Y^{\text{CW}} + (1 - \tau)q\right) = (1 - \alpha)^2, \tag{33}$$

*where $q := q(\mathbb{P})$ is the (exact) $(1 - \alpha)$-quantile of $S$ and for $y \in \mathcal{Y}$, $q_y^{\text{CW}} := q_y^{\text{CW}}(\mathbb{P})$ is the (exact) $(1 - \alpha)$-quantile of $S|Y = y$.*

*Proof.* Let us define explicitly the distributions $\mathbb{P}$ for which the upper bound is achieved. The distribution of $(S, Y) \sim \mathbb{P}$ is defined by:

$$Y \sim (1 - \alpha)\delta_{y_0} + \alpha\delta_{y_1}$$
$$S|(Y = y_0) \sim (1 - \alpha)\delta_0 + \alpha\delta_{1/2}, \quad S|(Y = y_1) \sim \delta_1,$$

for $y_0 \neq y_1 \in \mathcal{Y}$. Then $q = 1/2$, $q_{y_0}^{\text{CW}} = 0$ and $q_{y_1}^{\text{CW}} = 1$. It follows that, for $\tau \in (0, 1)$,

$$
\begin{aligned}
\mathbb{P}\left(S \leq \tau q_Y^{\text{CW}} + (1 - \tau)q\right) &= \mathbb{P}\left(S \leq \tau q_{y_0}^{\text{CW}} + (1 - \tau)q | Y = y_0\right)(1 - \alpha) \\
&\quad + \mathbb{P}\left(S \leq \tau q_{y_1}^{\text{CW}} + (1 - \tau)q | Y = y_1\right)\alpha \\
&= \mathbb{P}\left(S \leq (1 - \tau)/2 | Y = y_0\right)(1 - \alpha) \\
&\quad + \mathbb{P}\left(S \leq \tau + (1 - \tau)/2 | Y = y_1\right)\alpha.
\end{aligned}
$$

By plugging in the exact distributions of $S|Y = y_0$ and $S|Y = y_1$, we get

$$\mathbb{P}\left(S \leq \tau q_Y^{\text{CW}} + (1 - \tau)q\right) = \mathbb{P}\left(S = 0 | Y = y_0\right)(1 - \alpha) + 0 = (1 - \alpha)^2,$$

where we have used that $(1 - \tau)/2 < 1/2$ and $\tau + (1 - \tau)/2 < 1$. $\qquad\square$

## B.4 RAW FUZZY RECOVERS STANDARD AND CLASSWISE CP

We first more formally state the result of Proposition 4.

**Proposition 7.** *Let* $\alpha \in [0, 1]$ *and assume* $\alpha$ *satisfies* $m\alpha \notin \mathbb{N}$ *for all* $m \in [n + 1]$.[3] *Also assume* $\Pi : \mathcal{Y} \to \Lambda$ *maps each class to a unique point.*

*If, for all* $u, v \in \Lambda$ *with* $u \neq v$, *the kernel* $h$ *satisfies*

$$h_\sigma(u, v) \to 0 \text{ as } \sigma \to 0 \qquad \text{and} \qquad h_\sigma(u, v) \to h_\sigma(u, u) \text{ as } \sigma \to \infty,$$

*then, for sufficiently small* $\sigma$, *for any* $x \in \mathcal{X}$ *and calibration set, we have*

$$\mathcal{C}_{\text{RAWFUZZY}}(x) = \mathcal{C}_{\text{CLASSWISE}}(x),$$

*and, for sufficiently large* $\sigma$, *for any* $x \in \mathcal{X}$ *and calibration set, we have*

$$C_{\text{RAWFUZZY}}(x) = \mathcal{C}_{\text{STANDARD}}(x).$$

*Proof.* Let us first recall that $\mathcal{C}_{\text{RAWFUZZY}}(X) = \mathcal{C}_{\text{LW}}(X; w_{\text{FUZZY}})$. In the rest of the proof, we will write $w_\sigma$ to refer to the weighting function $w_{\text{FUZZY}}$ with bandwidth $\sigma > 0$. As described in (17), the weighting function $w_\sigma$ in the second argument of $\mathcal{C}_{\text{LW}}(\cdot)$ is used to obtain the class-specific thresholds $\hat{q}_y^{w_\sigma}$, and the label-weighted conformal prediction set is constructed as $\{y \in \mathcal{Y} : s(x, y) \leq \hat{q}_y^{w_\sigma}\}$. For all $y \in \mathcal{Y}$, we will show that as $\sigma \to 0$, we eventually have $\hat{q}_y^{w_\sigma} = \hat{q}$, and as $\sigma \to \infty$, we eventually have $\hat{q}_y^{w_\sigma} = \hat{q}_y^{\text{CW}}$, where $\hat{q}$ and $\hat{q}_y^{\text{CW}}$ are the STANDARD and CLASSWISE quantiles from (4) and (6).

To show that a scalar $x$ is equal to the $1 - \alpha$ quantile of a distribution $Q$, we will show

- *Step 1:* $\mathbb{P}_{X \sim Q}(X \leq x) \geq 1 - \alpha$.

- *Step 2:* For any $t < x$, we have $\mathbb{P}_{X \sim Q}(X \leq t) < 1 - \alpha$.

In the remainder of this proof, we will use two ways of indexing the calibration scores. First, for $i \in [n]$, we let $S_i := s(X_i, Y_i)$. Second, we index the calibration scores by class: for each class $y \in \mathcal{Y}$, we use $S_i^y$ for $i \in [n_y]$ to denote the calibration scores for class $y$. Before beginning, we also note that the assumption that $m\alpha \notin \mathbb{N}$ for all $m \in [n + 1]$ ensures $\lceil (n_y + 1)(1 - \alpha) \rceil > (n_y + 1)(1 - \alpha) > \lfloor (n_y + 1)(1 - \alpha) \rfloor$ for all $y \in \mathcal{Y}$. This will be used below.

---

[3]If the desired miscoverage level $\alpha$ does not satisfy this assumption, note that there exists an $\tilde{\alpha}$ that is infinitesimally close to $\alpha$ that does satisfy the assumption. Stated formally, for any $\alpha \in [0, 1]$ and any $\epsilon > 0$, there exists $\tilde{\alpha}$ such that $|\tilde{\alpha} - \alpha| \leq \epsilon$ and $m\tilde{\alpha} \notin \mathbb{N}$ for all $m \in [n + 1]$. This is due to the density of irrationals in real numbers and the fact that any irrational $\tilde{\alpha}$ satisfies the assumption.

CONVERGENCE TO $\mathcal{C}_{\text{CLASSWISE}}$ FOR SMALL $\sigma$. We will show that for sufficiently small $\sigma$, we have $\hat{q}_y^{\text{CW}} = \hat{q}_y^{w_{\text{FUZZY}}}$ for all $y \in \mathcal{Y}$, where

$$\hat{q}_y^{w_{\text{FUZZY}}} = \text{Quantile}_{1-\alpha}\Big(\underbrace{\frac{\sum_{i=1}^n w_\sigma(Y_i, y)\delta_{S_i} + w_\sigma(y, y)\delta_\infty}{\sum_{i=1}^n w_\sigma(Y_i, y) + w_\sigma(y, y)}}_{:=Q_y}\Big).$$

In other words, $\hat{q}_y^{w_{\text{FUZZY}}}$ is the $1 - \alpha$ quantile of $Q_y$. We now apply the two-step procedure outlined above to show that $\hat{q}_y^{\text{CW}}$ is equal to the $1 - \alpha$ quantile of $Q_y$ when $\sigma$ is sufficiently small.

*Step 1.* We begin by observing

$$\begin{aligned}
\mathbb{P}_{S \sim Q_y}\left(S \le \hat{q}_y^{\text{CW}}\right) &= \frac{\sum_{i=1}^n w_\sigma(Y_i, y)\mathbb{1}\{S_i \le \hat{q}_y^{\text{CW}}\}}{\sum_{i=1}^n w_\sigma(Y_i, y) + w_\sigma(y, y)} \\
&= \frac{w_\sigma(y, y)\sum_{i=1}^{n_y}\mathbb{1}\{S_i^y \le \hat{q}_y^{\text{CW}}\} + \sum_{z \in \mathcal{Y}\backslash\{y\}} w_\sigma(z, y)\sum_{i=1}^{n_z}\mathbb{1}\{S_i^z \le \hat{q}_y^{\text{CW}}\}}{(n_y + 1)w_\sigma(y, y) + \sum_{z \in \mathcal{Y}\backslash\{y\}} n_z w_\sigma(z, y)} \\
&\ge \frac{w_\sigma(y, y)\lceil(1 - \alpha)(n_y + 1)\rceil}{(n_y + 1)w_\sigma(y, y) + \sum_{z \in \mathcal{Y}\backslash\{y\}} n_z w_\sigma(z, y)},
\end{aligned}\tag{34}$$

where we have used for the last inequality the definition of the CLASSWISE quantile and have lower bounded the second term of the numerator by 0. By assumption, we know that $w_\sigma(z, y) \to 0$ as $\sigma \to 0$ for all classes $z \ne y$. Thus, for $\sigma$ small enough, we have

$$\sum_{z \in \mathcal{Y}\backslash\{y\}} n_z w_\sigma(z, y) \le \frac{w_\sigma(y, y)}{1 - \alpha}\Big(\lceil(n_y + 1)(1 - \alpha)\rceil - (n_y + 1)(1 - \alpha)\Big).$$

For such $\sigma$, (34) becomes

$$\mathbb{P}_{S \sim Q_y}\left(S \le \hat{q}_y^{\text{CW}}\right) \ge 1 - \alpha,$$

which implies that $\hat{q}_y^{\text{CW}} \ge \hat{q}_y^{w_{\text{FUZZY}}}$.

*Step 2.* We now show the converse inequality. Given any $t < \hat{q}_y^{\text{CW}}$, we want to show $\mathbb{P}_{S \sim Q_y}(S \le t) < 1 - \alpha$. Similar to above, we begin by writing

$$\begin{aligned}
\mathbb{P}_{S \sim Q_y}(S \le t) &= \frac{w_\sigma(y, y)\sum_{i=1}^{n_y}\mathbb{1}\{S_i^y \le t\} + \sum_{z \in \mathcal{Y}\backslash\{y\}} w_\sigma(z, y)\sum_{i=1}^{n_z}\mathbb{1}\{S_i^z \le t\}}{(n_y + 1)w_\sigma(y, y) + \sum_{z \in \mathcal{Y}\backslash\{y\}} n_z w_\sigma(z, y)} \\
&\le \frac{w_\sigma(y, y)\lfloor(n_y + 1)(1 - \alpha)\rfloor + \sum_{z \in \mathcal{Y}\backslash\{y\}} n_z w_\sigma(z, y)}{(n_y + 1)w_\sigma(y, y)}.
\end{aligned}\tag{35}$$

The inequality is obtained by *(i)* removing a positive term from the denominator and *(ii)* observing that if $t < \hat{q}_y^{\text{CW}}$, then at most $\lfloor(n_y+1)(1-\alpha)\rfloor$ scores of class $y$ are smaller or equal to $t$. Otherwise, $t$ would be higher than the class-conditional quantile $\hat{q}_y^{\text{CW}}$. By assumption, we know that $w_\sigma(z, y) \to 0$ as $\sigma \to 0$ for all classes $z \ne y$. Thus, for $\sigma$ small enough, we have

$$\sum_{z \in \mathcal{Y}\backslash\{y\}} n_z w_\sigma(z, y) < w_\sigma(y, y)\Big((1 - \alpha)(n_y + 1) - \lfloor(n_y + 1)(1 - \alpha)\rfloor\Big).$$

For such $\sigma$, (35) becomes

$$P_{S \sim Q_y}(S \le t) < 1 - \alpha.$$

Thus, $\hat{q}_y^{\text{CW}} \le \hat{q}_y^{w_{\text{FUZZY}}}$. Together, Step 1 and Step 2 tell us that for $\sigma$ small enough, we have $\hat{q}_y^{w_{\text{FUZZY}}} = \hat{q}_y^{\text{CW}}$, which concludes the proof for the convergence of $\mathcal{C}_{\text{FUZZY}}$ to $\mathcal{C}_{\text{CLASSWISE}}$.

CONVERGENCE TO $\mathcal{C}_{\text{STANDARD}}$ FOR LARGE $\sigma$. Recall that we assume as $\sigma \to \infty$, we have $w_\sigma(z, y) \to w_\sigma(y, y)$ for all $y, z \in \mathcal{Y}$. Then for any $\varepsilon > 0$, there exists $\sigma$ large enough such that

$$\max_{y, z \in \mathcal{Y}}\Big|\frac{w_\sigma(z, y)}{w_\sigma(y, y)} - 1\Big| \le \varepsilon \tag{36}$$

*Step 1.* As in the previous case, we write

$$
\begin{aligned}
\mathbb{P}_{S \sim Q_y}(S \leq \hat{q}) &= \frac{\sum_{i=1}^{n} w_\sigma(Y_i, y) \mathbb{1}\{S_i \leq \hat{q}\}}{\sum_{i=1}^{n} w_\sigma(Y_i, y) + w_\sigma(y, y)} \\
&= \frac{\sum_{z \in \mathcal{Y}} w_\sigma(z, y) \sum_{i=1}^{n_z} \mathbb{1}\{S_i^z \leq \hat{q}\}}{\sum_{z \in \mathcal{Y}} n_z w_\sigma(z, y) + w_\sigma(y, y)} \\
&\geq \frac{w_\sigma(y, y)(1 - \varepsilon) \sum_{i=1}^{n} \mathbb{1}\{S_i \leq \hat{q}\}}{(n(1 + \varepsilon) + 1) w_\sigma(y, y)} \\
&\geq \frac{1 - \varepsilon}{1 + \varepsilon} \frac{\lceil (n+1)(1 - \alpha) \rceil}{n + 1}.
\end{aligned}
$$

The first inequality is obtained by lower bounding $w_\sigma(z, y)$ by $(1 - \varepsilon) w_\sigma(y, y)$ in the numerator and upper bounding $w_\sigma(z, y)$ by $(1 + \varepsilon) w_\sigma(y, y)$ in the denominator. The last inequality comes from the fact that $\hat{q}$ is the empirical quantile of the scores. By choosing $\varepsilon$ small enough, specifically,

$$
\varepsilon \leq \frac{\lceil (n+1)(1 - \alpha) \rceil - (n+1)(1 - \alpha)}{\lceil (n+1)(1 - \alpha) \rceil + (n+1)(1 - \alpha)},
$$

we get that for $\sigma$ large enough, $\mathbb{P}_{S \sim Q_y}(S \leq \hat{q}) \geq 1 - \alpha$. Thus for such $\sigma$, we have $\hat{q} \geq \hat{q}_y^{w_{\text{FUZZY}}}$ for all $y \in \mathcal{Y}$.

*Step 2.* We now show the converse inequality. Given any $t < \hat{q}$, we want to show $\mathbb{P}_{S \sim Q_y}(S \leq t) < 1 - \alpha$. We start by observing

$$
\begin{aligned}
\mathbb{P}_{S \sim Q_y}(S \leq t) &= \frac{\sum_{z \in \mathcal{Y}} w_\sigma(z, y) \sum_{i=1}^{n_z} \mathbb{1}\{S_i^z \leq t\}}{\sum_{z \in \mathcal{Y}} n_z w_\sigma(z, y) + w_\sigma(y, y)} \\
&\leq \frac{1 + \varepsilon}{1 - \varepsilon} \frac{\sum_{i=1}^{n} \mathbb{1}\{S_i \leq t\}}{n + 1} \leq \frac{1 + \varepsilon}{1 - \varepsilon} \frac{\lfloor (n+1)(1 - \alpha) \rfloor}{n + 1}.
\end{aligned}
$$

We have used (36) again for the first inequality. For the last inequality, since $t < \hat{q}$, there are at most $\lfloor (n+1)(1 - \alpha) \rfloor$ scores which are smaller or equal $t$. Otherwise, $t$ would be higher than $\hat{q}$. Choosing $\varepsilon$ small enough such that

$$
\frac{1 + \varepsilon}{1 - \varepsilon} \frac{\lfloor (n+1)(1 - \alpha) \rfloor}{n + 1} < 1 - \alpha
$$

yields $\mathbb{P}_{S \sim Q_y}(S \leq t) \leq 1 - \alpha$, which implies that $\hat{q} \leq \hat{q}_y^{w_{\text{FUZZY}}}$. Combining the results from Step 1 and Step 2, we get that $\hat{q}_y^{w_{\text{FUZZY}}} = \hat{q}$ for $\sigma$ large enough, so $\mathcal{C}_{\text{FUZZY}}(X) \equiv \mathcal{C}_{\text{STANDARD}}(X)$.

INFINITE QUANTILES CASE. The above proof assumes that the STANDARD and CLASSWISE quantiles are not infinite. In the infinite case where $\hat{q} = \infty$ or $\hat{q}_y^{\text{CW}} = \infty$, we directly get Step 1 as then $\mathbb{P}_{S \sim Q_y}(S \leq q) = 1 \geq 1 - \alpha$ for $q \in \{\hat{q}, \hat{q}_y^{\text{CW}}\}$. The rest of the proof remains similar to before.

$\square$

## B.5 RECONFORMALIZATION OF RAW FUZZY

*Proof of Proposition 5.* This follows from the exchangeability of the held-out set $\mathcal{D}_{\text{hold}}$ and test point $(X_{n+1}, Y_{n+1})$. We first link the set $\mathcal{C}_{\text{RAWFUZZY}}$ to the score function $\tilde{s}$. For $x \in \mathcal{X}$, $y \in \mathcal{Y}$, let $\omega_y^y = w_{\text{FUZZY}}(y, y) / \left( \sum_{i=1}^{n} w_{\text{FUZZY}}(Y_i, y) + w_{\text{FUZZY}}(y, y) \right)$, then:

$$
y \in \mathcal{C}_{\text{RAWFUZZY}}(x) \iff s(x, y) \leq \text{Quantile}_{1-\alpha} \left( \sum_{i=1}^{n} w_i^y \delta_{s(X_i, Y_i)} + w_y^y \delta_\infty \right)
$$

$$
\iff \sum_{i=1}^{n} w_i^y \mathbb{1}\{s(X_i, Y_i) < s(x, y)\} < 1 - \alpha \iff \tilde{s}(x, y) < 1 - \alpha. \quad (37)
$$

By exchangeability, the set $\left\{ y : \tilde{s}(x, y) \leq \text{Quantile}_{1-\alpha} \left( \frac{1}{m+1} \sum_{i=1}^{m} \delta_{\tilde{s}(X_i^{\mathcal{H}}, Y_i^{\mathcal{H}})} + \frac{1}{m+1} \delta_\infty \right) \right\}$ has a marginal coverage of $1 - \alpha$. $\square$

We now briefly present a possible adaptation of the reconformalization step for FUZZY CP that avoids the need for an additional dataset $\mathcal{D}_{\text{hold}}$. To do so, we adapt the score $\tilde{s}$ from (19) to recalibrate the weighted quantile with the same dataset $\mathcal{D}_{\text{cal}}$ using full conformal techniques (Vovk et al., 2005).

**Proposition 8** (Reconformalization with $\mathcal{D}_{\text{cal}}$). *Let us define for $(x, y)$ and a finite subset $\mathcal{D}$ of $\mathcal{X} \times \mathcal{Y}$ the score*

$$\tilde{s}_{\text{full}}\big((x, y), \mathcal{D}\big) := \frac{\sum_{(x', y') \in \mathcal{D}} w_{\text{FUZZY}}(y', y) \mathbb{1}\{s(x', y') < s(x, y)\}}{\sum_{(x', y') \in \mathcal{D}} w_{\text{FUZZY}}(y', y)} \,.$$

*Given a calibration set $\mathcal{D}_{\text{cal}} = \{(X_i, Y_i)\}_{i=1}^n$, let $S_i(x, y) = \tilde{s}_{\text{full}}\big((X_i, Y_i), \mathcal{D}_{\text{cal}} \cup (x, y)\big)$. Then the set*

$$\mathcal{C}_{\text{FULL}}(x) = \left\{ y \in \mathcal{Y} : \tilde{s}_{\text{full}}\big((x, y), \mathcal{D}_{\text{cal}} \cup (x, y)\big) \leq \text{Quantile}_{1-\alpha}\Big(\frac{1}{n+1} \sum_{i=1}^{n+1} \delta_{S_i(x,y)} + \frac{1}{n+1} \delta_\infty\Big) \right\}$$

*has marginal coverage of $1 - \alpha$ for the test point, as long as it is exchangeable with $\mathcal{D}_{\text{cal}}$.*

*Proof.* As long as the test point $(X_{n+1}, Y_{n+1})$ and the points in $\mathcal{D}_{\text{cal}}$ are exchangeable, the scores $S_i(X_{n+1}, Y_{n+1})$ and $S_{n+1}(X_{n+1}, Y_{n+1}) := \tilde{s}_{\text{full}}\big((X_{n+1}, Y_{n+1}), \mathcal{D}_{\text{cal}} \cup (X_{n+1}, Y_{n+1})\big)$ are also exchangeable, which yields the marginal validity of $\mathcal{C}_{\text{FULL}}$. $\square$

In cases where the amount of calibration data is limited, this full-conformal adaptation allows us to avoid data-splitting. It is also computationally feasible in such scenarios, as the burden of recalculating the ensemble thresholds for each test point is tolerable when $\mathcal{D}_{\text{cal}}$ is small.

## C EXPERIMENT DETAILS

### C.1 DATASET PREPARATION

**Overview.** Our dataset preparation has two steps. We first identify or create `train`/`val`/`test` splits to replicate the standard machine learning pipeline. After obtaining `train`/`val`/`test` splits, we further split `val` into two subsets: the first for selecting the number of epochs (containing a random 30% of `val`) and the second for use as the calibration set $\mathcal{D}_{\text{cal}}$ (containing the remaining 70%). This split is necessary to ensure that $\mathcal{D}_{\text{cal}}$ is exchangeable with the test points; if we reuse `val` for both epoch selection and calibration, this would violate exchangeability.

We now describe the `train`/`val`/`test` splits for each of the datasets we use.

*Pl@ntNet-300K.* We use the provided `train`/`val`/`test` splits of Pl@ntNet-300K.[4] The creation of the dataset is described in Garcin et al. (2022).

*iNaturalist-2018.* We use the 2018 version of iNaturalist[5] because it has the most "natural" class distribution (i.e., not truncated). Unfortunately, the provided `train`/`val`/`test` splits are not well-suited to applying and evaluating conformal prediction methods for two reasons: First, the provided `test` set is not labeled and cannot be used for evaluation purposes. Second, the provided `val` set is class-balanced (with three examples per class), which means that it is not a representative sample of the test distribution. If we used this validation set as our conformal calibration set, this would violate the key assumption that the calibration points are exchangeable with the test points. To remedy these two problems, we create our own `train`/`val`/`test` splits where all splits have the same distribution. Specifically, we aggregate the `train` and `val` data, then for each class randomly select 80% of examples to put in our `train`, and then put 10% each in our `val` and `test`.

*Truncated versions.* A key challenge of the long-tailed datasets we work with is that their test sets are also long-tailed, which hinders reliable class-conditional evaluation for rare classes. To address this, we create truncated versions of the datasets by removing classes with fewer than 101 examples, resulting in 330 Pl@ntNet-300K classes and 857 iNaturalist classes. For each remaining class, we assign 100 examples to `test`, then divide the remainder 90%/10% into `train`/`val`. For rare classes, we prioritize training set allocation, which may lead to having zero calibration examples.

---

[4] https://github.com/plantnet/PlantNet-300K
[5] https://github.com/visipedia/inat_comp/tree/master/2018

Specifically, to obtain *Pl@ntNet-300K-truncated*, we apply the above procedure to the combined `train`, `val`, and `test` splits of Pl@ntNet-300K. To obtain *iNaturalist-2018-truncated*, we apply the procedure to the combined `train` and `val` splits of the original iNaturalist dataset.[6]

**Classes with zero calibration examples.**    Due to randomness in the data splitting, some classes end up with no calibration examples: 105 (out of 1081) in Pl@ntNet-300K, 606 (out of 8142) in iNaturalist-2018, 12 (out of 330) in truncated Pl@ntNet-300K, and 47 (out of 857) in truncated iNaturalist-2018.

## C.2   MODEL TRAINING

After following the procedures described in Appendix C.1 to obtain `train`, `val`, and `test` splits of each dataset, we train a ResNet-50 (He et al., 2016) initialized to ImageNet pretrained weights for 20 epochs using a learning rate of 0.0001. We use `train` for training the neural network and the randomly selected 30% of `val` for computing the validation accuracy. We then select the epoch number that results in the highest validation accuracy (up to 20 epochs).

## C.3   COMPUTATIONAL RESOURCES

The system we use is equipped with a 4x Intel Xeon Gold 6142 (64 cores/128 threads total @ 2.6-3.7 GHz, 88MB L3 cache) while the GPUs are 2x NVIDIA A10 (24GB VRAM each) and 2x NVIDIA RTX 2080 Ti (11GB VRAM each), for a total of 70GB GPU VRAM.

# D   ADDITIONAL EXPERIMENTAL RESULTS

**Overview.**    In this section, we extend the experimental results from the main paper in the following ways:

1. We include additional methods: To ensure that we are evaluating fairly against existing procedures, we include two additional baseline methods. The first is called EXACT CLASS-WISE, a randomized version of classwise conformal that is designed to achieve *exact* (rather than *at least*) $1 - \alpha$ coverage, as described in Appendix C.3 of Ding et al. (2023). The second is rank calibrated class-conditional conformal prediction (RC3P), proposed by Shi et al. (2024). We also include the Raw FUZZY and FUZZY methods we propose in Appendix A.

2. We evaluate on the truncated versions of Pl@ntNet-300K and iNaturalist-2018, as described in Appendix C.1.

In Appendix D.1, we provide additional results to complement Figure 3 in the main paper. In Appendix D.2, we provide results for additional methods. In Appendix D.3, we recreate Figure 3 using a ResNet-50 trained using focal loss rather than cross-entropy loss. In Appendix D.4, we do a case study of the implications of our methods for plant identification. In Appendix D.5, we visualize the score thresholds we obtain from our methods to demonstrate how they interpolate between the STANDARD and CLASSWISE thresholds.

## D.1   RESULTS FOR MAIN METHODS

To aid interpretation of Figure 3, we extract the metric values for select methods for $\alpha = 0.1$ and present them in Table 3 for Pl@ntNet-300K and Table 4 for iNaturalist-2018. We also include a comprehensive visualization of all methods on all datasets, including the truncated versions, in Figure 7. We remark that the two existing conformal prediction methods for many-class classification perform poorly in the long-tailed setting. The CLUSTERED CP method from Ding et al. (2023) defaults to the score threshold from STANDARD CP for classes that have insufficient examples to confidently assign to a cluster, resulting in poor class-conditional coverage. Meanwhile, the RC3P method from Shi et al. (2024) suffers from a similar data splitting problem as CLASSWISE CP, as it strongly relies on the estimation of the conditional score quantile, which can be very bad for classes with few calibration examples. Furthermore, it requires the estimation of the conditional top-k error for all classes, which is also hard when classes have few calibration examples. Moreover, as these

---

[6]For truncated datasets, we must compute marginal metrics differently when doing evaluation: due to the uniform class distribution of the `test` splits of these datasets, a simple average over all test examples does not reflect marginal performance on the true distribution. We estimate the marginal coverage as $\sum_{y \in \mathcal{Y}} \hat{p}(y)\hat{c}_y$ where $\hat{p}(y)$ is estimated using `train` and $\hat{c}_y$ is the empirical class-conditional coverage as defined in (16). A similar weighting procedure is used to estimate the average set size.

quantities are estimated on the calibration set, the exchangeability assumption is violated which explains why the marginal coverage is sometimes below $1 - \alpha$. The methods we propose strictly dominate RC3P; for a given average set size, we get much better class-conditional coverage.

Table 3: Set size and coverage metrics for Pl@ntNet-300K using the $s_{\mathsf{softmax}}$ score and $\alpha = 0.1$. The arrows next to the coverage metric names indicate whether it is better for the metric to be smaller ($\downarrow$) or larger ($\uparrow$).

| Method | FracBelow50% $\downarrow$ | UnderCovGap $\downarrow$ | MacroCov $\uparrow$ | MarginalCov (desired $\geq 0.9$) | Avg. set size $\downarrow$ |
|---|---|---|---|---|---|
| STANDARD | 0.389 | 0.398 | 0.525 | 0.907 | 1.57 |
| CLASSWISE | 0.000 | 0.006 | 0.976 | 0.912 | 780.00 |
| CLUSTERED | 0.398 | 0.406 | 0.513 | 0.882 | 1.57 |
| STANDARD w. PAS | 0.167 | 0.193 | 0.755 | 0.902 | 2.57 |
| INTERP-Q ($\tau = 0.9$) | 0.248 | 0.265 | 0.671 | 0.901 | 2.24 |
| INTERP-Q ($\tau = 0.99$) | 0.151 | 0.168 | 0.785 | 0.905 | 3.95 |
| INTERP-Q ($\tau = 0.999$) | 0.098 | 0.109 | 0.856 | 0.908 | 7.58 |

Table 4: Set size and coverage metrics for iNaturalist-2018 using the $s_{\mathsf{softmax}}$ score and $\alpha = 0.1$.

| Method | FracBelow50% $\downarrow$ | UnderCovGap $\downarrow$ | MacroCov $\uparrow$ | MarginalCov (desired $\geq 0.9$) | Avg. set size $\downarrow$ |
|---|---|---|---|---|---|
| STANDARD | 0.058 | 0.116 | 0.849 | 0.902 | 10.9 |
| CLASSWISE | 0.000 | 0.002 | 0.992 | 0.954 | 7430.0 |
| CLUSTERED | 0.059 | 0.118 | 0.845 | 0.880 | 8.4 |
| STANDARD w. PAS | 0.042 | 0.093 | 0.875 | 0.900 | 11.3 |
| INTERP-Q ($\tau = 0.9$) | 0.034 | 0.081 | 0.891 | 0.907 | 16.8 |
| INTERP-Q ($\tau = 0.99$) | 0.020 | 0.055 | 0.924 | 0.923 | 31.1 |
| INTERP-Q ($\tau = 0.999$) | 0.010 | 0.037 | 0.947 | 0.934 | 55.8 |

## D.2 RESULTS FOR ADDITIONAL METHODS

**FUZZY with other class mappings.** Recall that in order to instantiate FUZZY CP, we must choose a mapping $\Pi$ that maps each class $y \in \mathcal{Y}$ to some low-dimensional space in which we can compute distances in a meaningful way. Depending on the setting, some mappings may work better than others. The mapping we proposed in the main paper, $\Pi_{\mathrm{prevalence}}$ is a simple mapping that works well in the long-tailed settings we considered. In this section, we present results for two additional mappings. We describe all three mappings here.

1. *Prevalence*: This is the mapping presented before, which maps each class to its popularity in the `train` set:

$$\Pi_{\mathrm{prevalence}}(y) = c n_y^{\mathrm{train}} + \varepsilon_y,$$

   where $\varepsilon_y \sim \mathrm{Unif}([-0.01, 0.01])$ independently for each for $y \in \mathcal{Y}$. We normalize using $c = 1/(\max_{y' \in \mathcal{Y}} n_{y'}^{\mathrm{train}})$ to ensure that $\Pi_{\mathrm{prevalence}}(y) \in [0, 1]$ so that the same bandwidth $\sigma$ has similar effects on different datasets.

2. *Random.* As a baseline, we try mapping each class to a random value. Specifically,

$$\Pi_{\mathrm{random}}(y) = U_y \qquad \text{where } U_y \overset{\mathrm{iid}}{\sim} \mathrm{Unif}([0, 1]) \text{ for } y \in \mathcal{Y}.$$

3. *Quantile.* Recall the intuition described in the last paragraph of Section 2, which says that we want a mapping that groups together classes with similar score distributions. To further develop this intuition, suppose that when computing $\hat{q}_y$, we assign non-zero weights only to classes with the same $1 - \alpha$ score quantile as class $y$. Taking the $1 - \alpha$ weighted quantile would then recover the $1 - \alpha$ quantile of class $y$ as the number of samples with non-zero weight grows. This is because the mixture of distributions with the same $1 - \alpha$

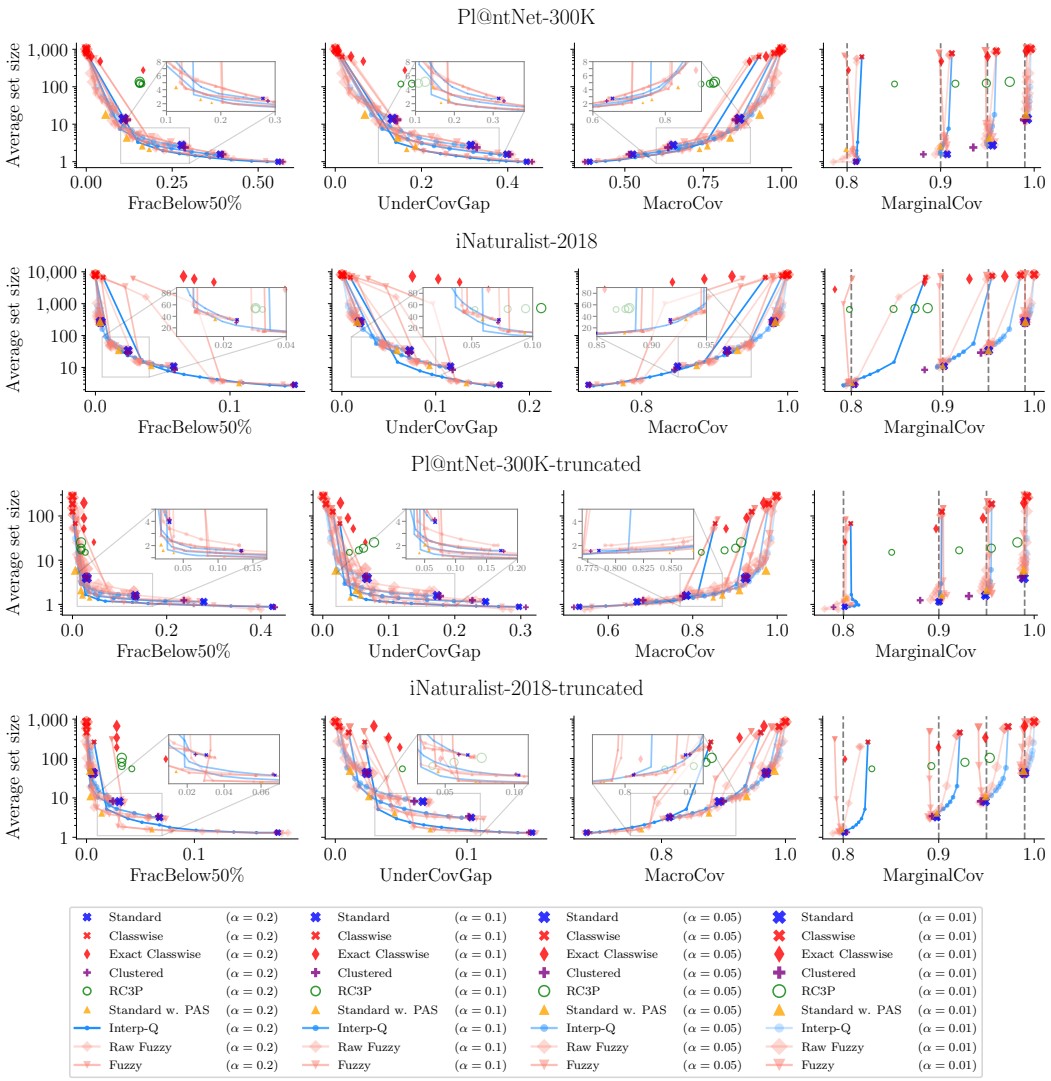

Figure 7: Average set size vs. FracBelow50%, UnderCovGap, MacroCov, and MarginalCov for various methods on full and truncated datasets. For methods with tunable parameters, lines are used to trace out the trade-off curve achieved by running the method with different values for a fixed $\alpha$. For FracBelow50% and UnderCovGap, it is better to be closer to the bottom left. For MacroCov, the bottom right is better. For MarginalCov, we want to be at the bottom and to the right of the dotted line at $1 - \alpha$ for the $\alpha$ at which the method is run.

quantile has that same $1 - \alpha$ quantile. Motivated by this idea, we map each class to the *linearly interpolated* empirical $1 - \alpha$ quantile of its scores in the calibration set. This is similar to $\hat{q}_y^{\mathrm{CW}}$ but not the same due to the linear interpolation. We chose to linearly interpolate to avoid the problem of rare classes being mapped to $\infty$, which happens if we apply no interpolation and directly apply $\mathrm{Quantile}_{1-\alpha}$ as it is defined in Section 1.2. Linear interpolation of quantiles is described in Definition 7 of Hyndman & Fan (1996) and is the default interpolation method implemented in `numpy.quantile()` (Harris et al., 2020). Given a finite set $A \subset \mathbb{R}$ and level $\tau \in [0, 1]$, let $\mathrm{LinQuantile}_\tau(A)$ denote the linearly interpolated $\tau$ quantile of the elements of $A$ or $s_{\max}$ if $A$ is empty, where $s_{\max} = \max_{i \in [n]} s(X_i, Y_i)$ is the maximum observed calibration score. The quantile projection is given by

$$\Pi_{\mathrm{quantile}}(y) = \mathrm{LinQuantile}_{1-\alpha}\big(\{s(X_i, Y_i)\}_{i \in \mathcal{I}_y}\big).$$

Results from applying FUZZY CP with $\Pi_{\text{random}}$ and $\Pi_{\text{quantile}}$ are shown in Figure 8. We observe that $\Pi_{\text{prevalence}}$ achieves a better trade-off than $\Pi_{\text{random}}$, which is expected, but it also performs better than $\Pi_{\text{quantile}}$, which is perhaps less expected. We believe that the reason that the quantile projection does not do well, despite being intuitively appealing, is that it is very sensitive to noise due to the low number of calibration examples per class. This likely causes classes that do not actually have similar score distributions to be mapped to similar values.

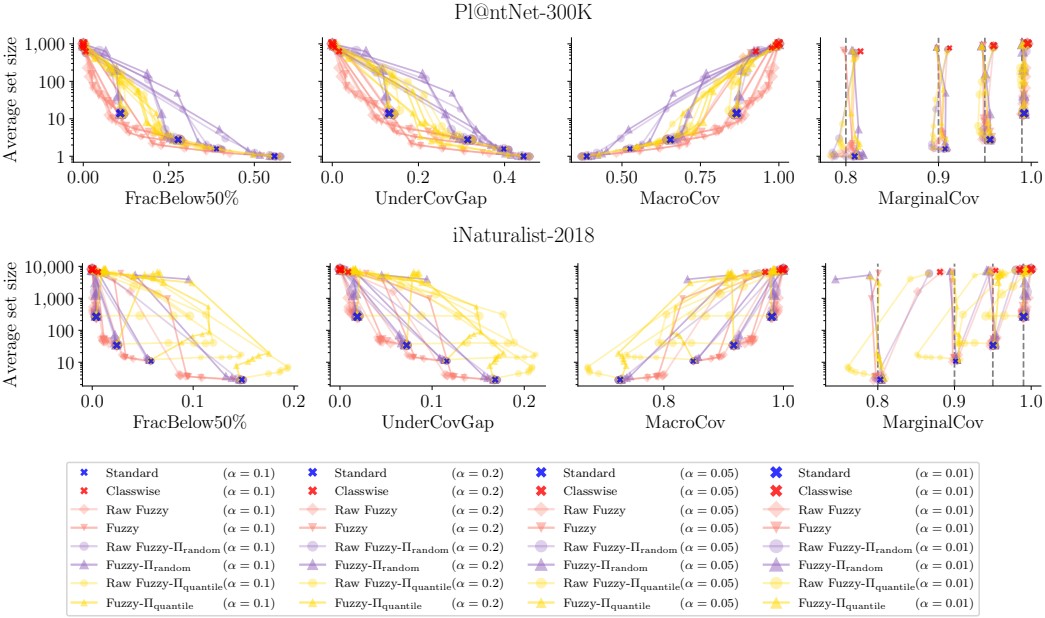

Figure 8: The performance of FUZZY CP under different class mappings $\Pi$.

**FUZZY and INTERP-Q with PAS.** Recall that we proposed two solution approaches in the main paper. One led to a conformal score function, PAS, and the other led to new procedures, INTERP-Q and FUZZY. One may wonder if combining the two approaches provides additional benefit over using just one, so we test this idea. In Figure 9, we plot the results of running FUZZY and INTERP-Q using PAS as the conformal score function. The thick blue ×'s correspond to STANDARD with PAS (previously shown as gold triangles in the main text), and we observe that the interpolation methods, FUZZY with PAS and INTERP-Q with PAS, do provide some additional benefit at appropriately chosen parameter values by more optimally trading off set size and UnderCovGap, while doing no worse than STANDARD with PAS in terms of other metrics.

### D.3 RESULTS USING FOCAL LOSS

To understand how our proposed conformal prediction methods can be combined with existing strategies for dealing with long-tailed distributions, we run additional experiments where we replace the cross-entropy loss with the focal loss in our ResNet-50 training. The focal loss was proposed by Lin et al. (2017) to improve model accuracy on rare classes by modifying the cross-entropy loss. We use the PyTorch implementation of focal loss from https://github.com/AdeelH/pytorch-multi-class-focal-loss with the default parameter values of $\gamma = 2$ and $\alpha = 1$. The results are shown in Figure 10 and are qualitatively similar to the cross-entropy results in Figure 3 in the main paper.

### D.4 PL@NTNET CASE STUDY

In this section, we highlight the importance of conformal prediction for settings like Pl@ntNet, a phone-based app that allows users to identify plants from images (Joly et al., 2014). Conformal prediction provides value to such applications by generating sets of possible labels instead of point predictions. These prediction sets should balance several desiderata:

1. *Marginal coverage.* For the general public, prediction sets improve the user experience relative to point predictions by offering multiple potential identifications, increasing the

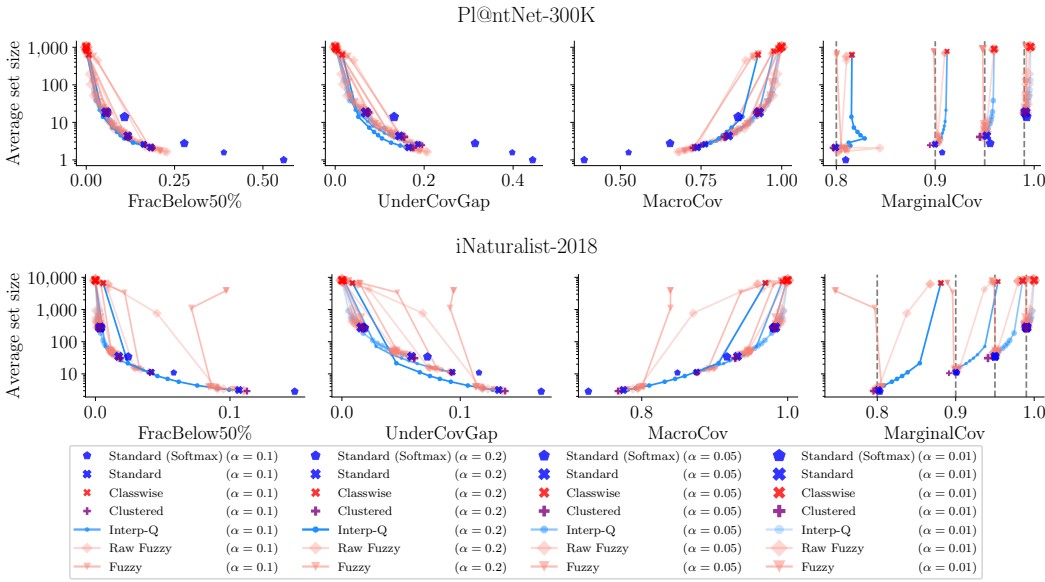

Figure 9: This plot is similar to Figure 3 except here we use PAS as the conformal score function instead of softmax. Note that the thick blue ×'s in this plot are equivalent to the gold triangles in Figure 3.

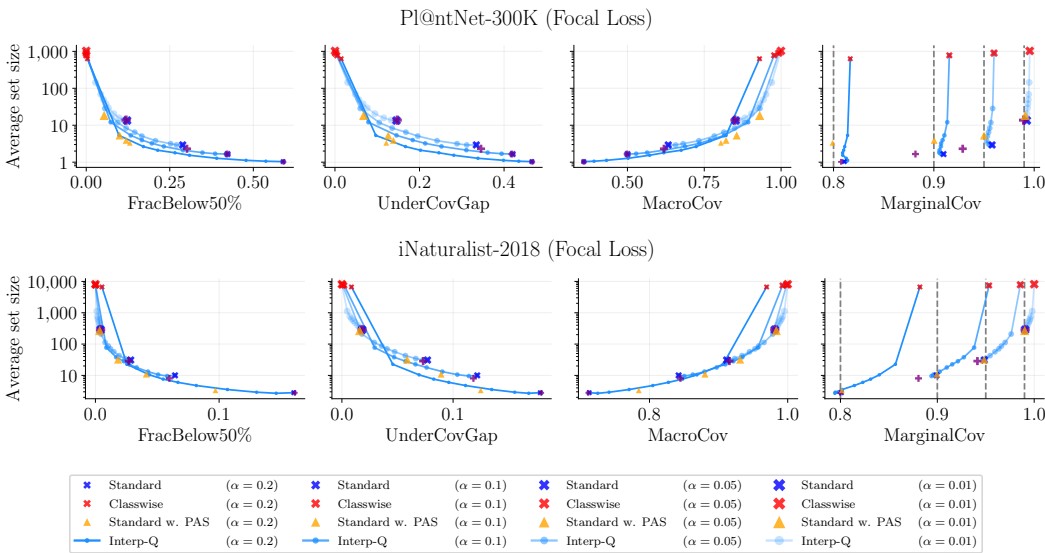

Figure 10: This figure is identical to Figure 3 except the base model is trained using focal loss (Lin et al., 2017).

likelihood of accurate identification even with ambiguous or low-quality images. In order for users to have a good chance of making the correct identification most of the time, we want the marginal coverage to be high.

2. *Class-conditional coverage (especially in the tail).* Ecologists would like to focus more on identifying (near) endangered or less commonly observed species for the purpose of scientific data collection. This calls for improving coverage of classes in the tail of the label distribution.

3. *Set size.* For both the general public and ecologists, maintaining reasonable set sizes is crucial, as sifting through large sets is impractical.

An additional challenge in plant identification is that visually similar species often have an imbalanced number of labeled images, with one species significantly more represented. Standard conformal methods can suffer from occlusion, where the dominant species overshadows the rarer one at prediction time, so that the classifier always assigns low probability to the rare class. If the rare species is never included in the prediction sets, this can lead to a vicious cycle of increasing imbalance, as users simply confirm the classifier's suggestions.

For this case study, we focus on comparing our proposed method STANDARD with PAS against STANDARD and CLASSWISE when run at the $\alpha = 0.1$ level. All methods have a marginal coverage guarantee, so we focus on comparing class-conditional coverage and set size between the methods. Figure 11 shows the class-conditional coverage and average set size for the three methods. Species that are considered endangered by the International Union for Conservation of Nature (IUCN) are highlighted in red. We observe that CLASSWISE results in huge prediction sets. On the other hand, STANDARD with PAS enhances the coverage of classes that have low coverage under STANDARD with softmax without producing huge sets. We provide visual examples of some endangered species from the Pl@ntNet-300K dataset in Figure 12.

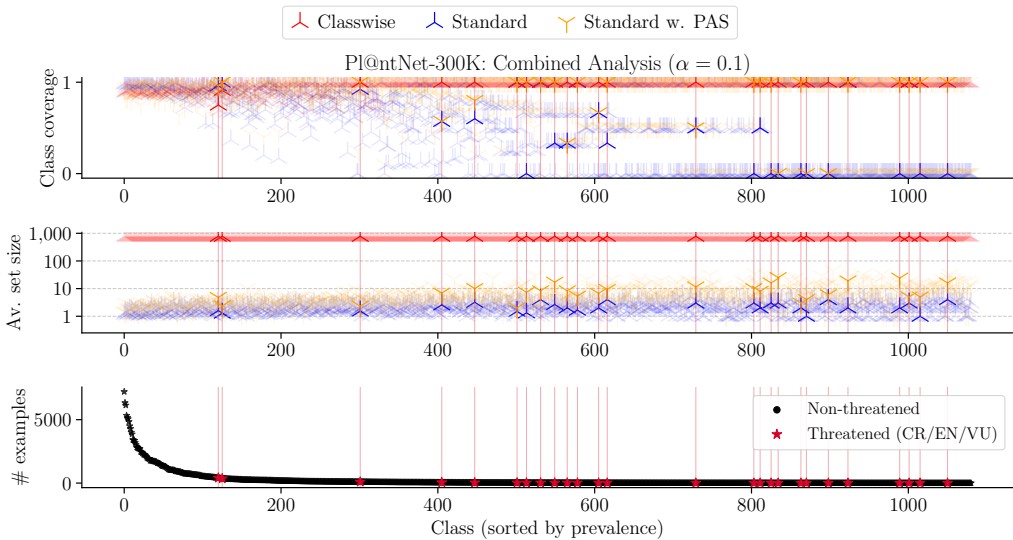

Figure 11: A detailed look at results by class on Pl@ntNet-300K dataset for three methods: STANDARD (with softmax), CLASSWISE (with softmax), and STANDARD with PAS. All methods are run at the $\alpha = 0.1$ level. Species are ordered according to the prevalence (computed on `train`), and the ones considered "threatened" according to the IUCN are highlighted in red.

### D.5 UNDERSTANDING MOVEMENTS IN $\hat{q}_y$

Figure 13 visualizes the $\hat{q}_y$ vectors that result from each of our methods. Recall that STANDARD has a single score threshold $\hat{q}$, which we plot as a horizontal line ($\hat{q}_y = \hat{q}$ for all classes $y$). As a reminder, a smaller value of $\hat{q}_y$ means that the class is less likely to be included in the prediction set, whereas classes with $\hat{q}_y = 1$ (or $\infty$) are always included in the prediction set. For STANDARD with PAS, we plot the *effective* $\hat{q}_y$ by observing that the class-uniform $\hat{q}$ threshold in terms of the PAS score implies classwise $\hat{q}_y$ thresholds in terms of the softmax score. Specifically, observe that

$$s_{\mathsf{PAS}}(x, y) \le \hat{q}$$

$$\iff -\frac{\hat{p}(y|x)}{\hat{p}(y)} \le \hat{q} \qquad \text{definition of } s_{\mathsf{PAS}}$$

$$\iff 1 - \hat{p}(y|x) \le 1 + \hat{q}\hat{p}(y)$$

$$\iff s_{\mathsf{softmax}}(x, y) \le 1 + \hat{q}\hat{p}(y)$$

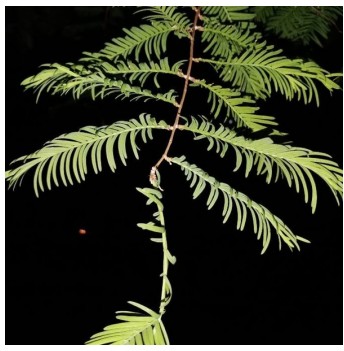 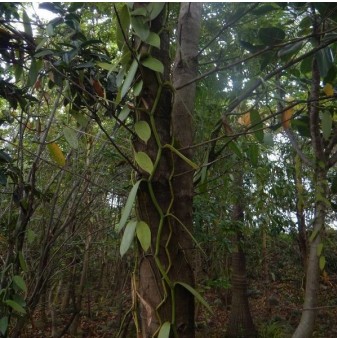

**Species**: *Metasequoia glyptostroboides Hu & W.C.Cheng*
**# of examples**: 410

| Method | Coverage | Size |
|---|---|---|
| Standard | 0.00 | 1.3 |
| Classwise | 1.00 | 781.3 |
| Std w. PAS | 1.00 | 7.3 |

**Species**: *Vanilla planifolia Jacks. ex Andrews*
**# of examples**: 35

| Method | Coverage | Size |
|---|---|---|
| Standard | 0.60 | 3.2 |
| Classwise | 1.00 | 782.2 |
| Std w. PAS | 0.80 | 10.0 |

**Species**: *Abeliophyllum distichum Nakai*
**# of examples**: 4

| Method | Coverage | Size |
|---|---|---|
| Standard | 0.00 | 4.0 |
| Classwise | 1.00 | 784.0 |
| Std w. PAS | 0.00 | 6.0 |

Figure 12: Examples of three species in Pl@ntNet-300K considered as "endangered" by the IUCN. Each table reports the empirical class-conditional coverage and the average size of the prediction sets when the given species is the true label for two baseline methods (STANDARD with softmax and CLASSWISE with softmax), as well as one of our proposed methods (STANDARD with PAS).

where $s_{\mathsf{softmax}}$ denotes the softmax score function described in Section 1.2. Thus, for a class $y \in \mathcal{Y}$, we refer to $1 + \hat{q}\hat{p}(y)$ as the effective $\hat{q}_y$ of STANDARD with PAS.

All of our methods are intended to "interpolate" between STANDARD (with softmax) and CLASSWISE (with softmax). INTERP-Q interpolates in a very literal sense by linearly interpolating $\hat{q}$ and $\hat{q}_y^{\mathsf{CW}}$. The other three methods appear to interpolate in an alternative geometry and allow the $\hat{q}_y$ for each class to be adjusted in a different way. These plots also reveal why CLASSWISE yields such large sets in this setting: there are many classes for which $\hat{q}_y^{\mathsf{CW}} = 1$ (or $\infty$), and all of these classes are always included in the prediction set.

### D.6 ADDITIONAL DECISION ACCURACY PLOTS

In the main text we present decision accuracy plots for only one of our methods, STANDARD with PAS. In Figure 14, we present results for our other method, INTERP-Q.

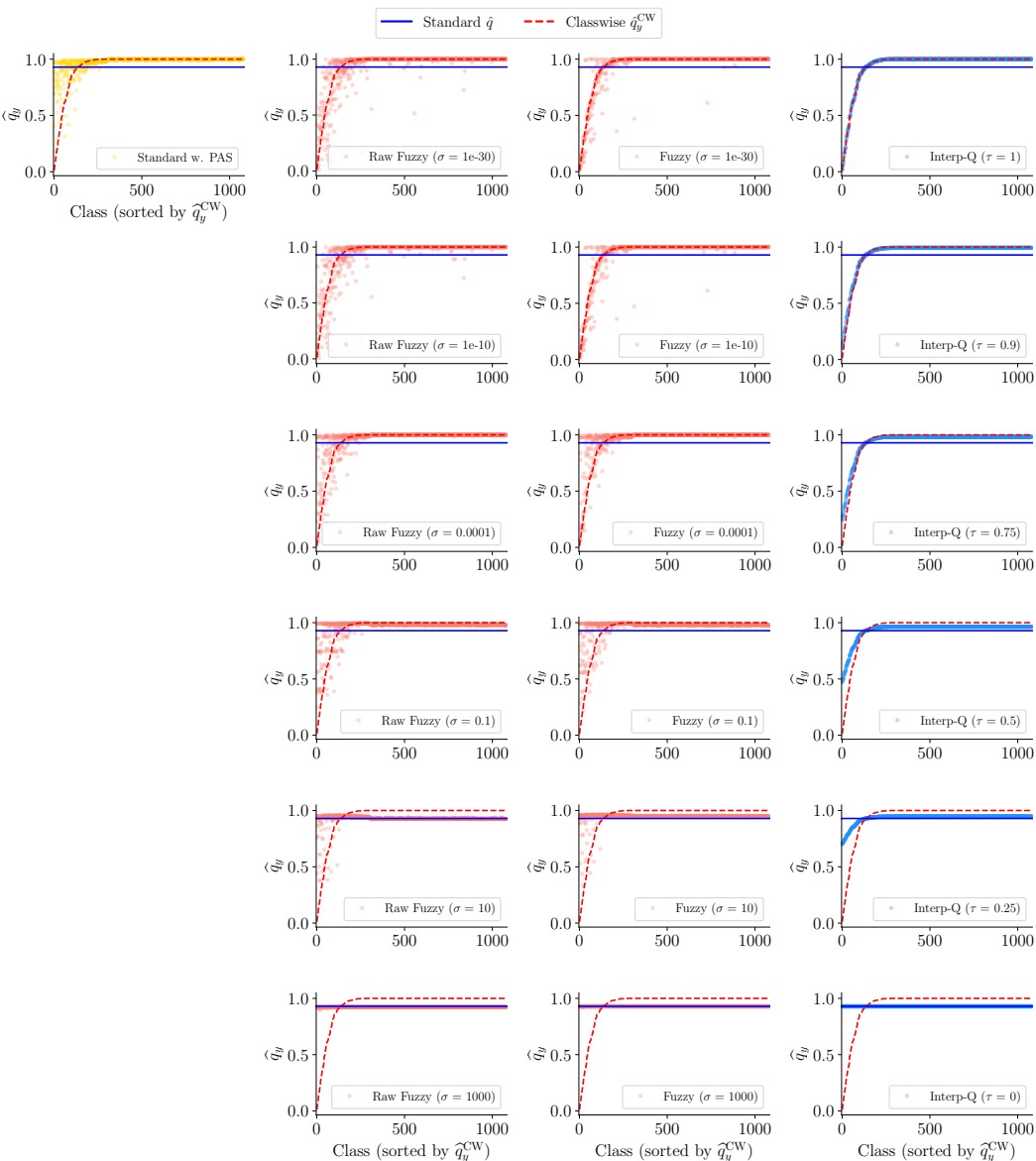

Figure 13: Score thresholds $\hat{q}_y$ of our proposed methods (STANDARD with PAS, RAW FUZZY, FUZZY, and INTERP-Q) on Pl@ntNet-300K for $\alpha = 0.1$. For visualization purposes, infinite values of $\hat{q}_y$ are replaced with one, the maximum possible softmax score value. Furthermore, for ease of comparison, we sort the classes in ascending value of the CLASSWISE thresholds $\hat{q}_y^{\text{CW}}$ (plotted as a dashed red line).

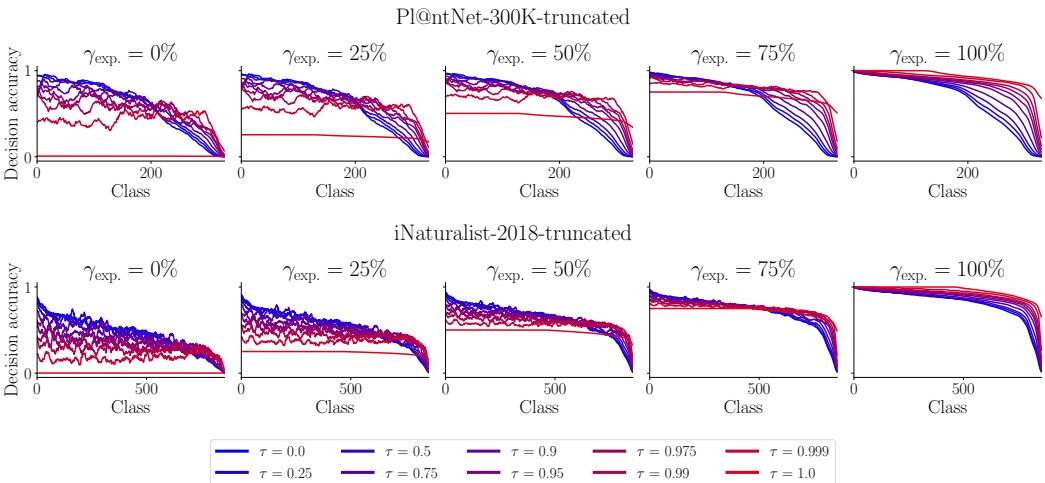

Figure 14: Class-conditional decision accuracies for a range of decision makers when presented with sets from INTERP-Q run with different values of $\tau$. Recall that $\tau = 0$ recovers STANDARD and $\tau = 1$ recovers CLASSWISE. Classes are ordered by decreasing decision accuracy of $H_{\text{expert}}$ under each method.

