# OpenReview forum: "Conformal Prediction for Long-Tailed Classification"
_ICLR.cc/2026/Conference — ICLR 2026 Poster_

### Official Review · Reviewer_mwex · 2025-10-30

**Soundness:** 3
**Presentation:** 4
**Contribution:** 3
**Rating:** 6
**Confidence:** 4

**Summary:**

This paper addresses the challenge of constructing useful prediction sets using conformal prediction (CP) in long-tailed classification settings, such as plant identification, where rare classes are underrepresented. Standard CP achieves marginal coverage but often fails to cover rare classes adequately, while class-conditional CP methods produce excessively large sets. The authors propose two approaches: (1) a new conformal score function called prevalence-adjusted softmax (PAS) and its weighted variant (WPAS) to target macro-coverage (the unweighted average of class-conditional coverages), derived from oracle-optimal sets that balance set size and coverage; (2) INTERP-Q, a simple interpolation between standard and classwise CP quantiles to trade off set size and class-conditional coverage. They evaluate these methods on long-tailed image datasets, demonstrating improved trade-offs compared to baselines.

**Strengths:**

The paper introduces novel conformal score functions (PAS and WPAS) specifically tailored for macro-coverage in long-tailed settings, grounded in theoretical derivations of oracle-optimal prediction sets. This extends prior work on CP beyond marginal or basic class-conditional guarantees, addressing a practical gap in applications like biodiversity monitoring where rare classes are critical. The paper is clear, with well-defined notation, algorithms, and metrics, making the contributions accessible.

**Weaknesses:**

The interpolation in INTERP-Q is simple but lacks deeper analysis of why linear weighting works well, and why it can only guarantee a conservative coverage bound ($1-2\alpha$) while being close to $1-\alpha$ in practice.

Additionally, comparisons could include more recent long-tail methods (e.g., Ding et al. (2023)).

**Questions:**

1. Can this method be combined with other scores, such as RAPS and SAPS?
2. In figure 2, there are cases where test label distribution is different from training and validation distribution, how this affects the results, especially considering PAS that is based on class normalization.
3. Can this method be extended to regression problems with class imbalance, where the continuous target variable has a highly skewed distribution with certain value ranges, much more frequent than others?

---

> ### Author Response · Authors · 2025-11-21
> **Addressing weaknesses & questions**
>
> **Weaknesses**
>
> a. The interpolation in INTERP-Q is simple but lacks deeper analysis of why linear weighting works well, and why it can only guarantee a conservative coverage bound (1-2alpha) while being close to 1-alpha in practice.
>
> > We have added an example to Appnedix B.3 where INTERP-Q achieves $1-2\alpha + \alpha^2$ coverage, demonstrating that our bound is almost tight. To elaborate on the intuition: Suppose that all of the class-conditional score thresholds are smaller than the marginal one. If this were true, interpolating between the marginal and class-conditional thresholds would result in coverage of at least $1-\alpha$, since the INTERP-Q prediction set would contain the marginal (Standard CP) prediction set. However, in pathological settings (such as in the newly added example), it is possible that this interpolation simultaneously reduces  the coverage of all classes, which lead to this lower bound of $1-2\alpha$.
> >
> > Note that the divergence between guaranteed coverage and observed coverage appear frequently in CP methods, e.g. for Jacknife+ or CV+ (Barber et al. 2020). The general point of view on this question is that some extreme cases are taken into account by the theory but do not happen in practice.
> >
> > As an additional remark, we highlight that INTERP-Q is not the only interpolation scheme we tried. As you suggest, it is not obvious that linear weighting is the best way to interpolate. In Appendix A, we present an additional method that interpolates in an alternative geometry that is based on some measure of "similarity" between classes. However, we found that this more sophisticated approach does not perform better than the simple INTERP-Q method empirically.
>
> b. Additionally, comparisons could include more recent long-tail methods (e.g., Ding et al. (2023)).
>
> >We agree that Ding et al. (2023) is a relevant baseline and it is indeed included in our paper, under the name "Clustered" (see Fig. 3, and additional results in appendix). The method essentially defaults to Standard CP when there are classes with insufficient examples to properly cluster, leading to poor class-conditional performance.
>
> **Questions**
>
> 1. Can this method be combined with other scores, such as RAPS and SAPS?
>
>
> > We proposed two methods: INTERP-Q and PAS. It is possible and straighforward to use INTERP-Q with any conformal score function (say APS, RAPS or SAPS), and our coverage guarantee would still hold. For prevalence-adjusted softmax, it is possible to apply the key idea of reweighting scores by the inverse of the prevalence, but it's unclear what problem you are solving theoretically when not using the softmax score as the initial score. The motivation for the PAS score relies on the base score (in our case, softmax) being an estimate of the conditional probability $p(y|x)$. Using a different base score might however still be of practical interest, for example in settings where both approximate X-conditional and class-conditional coverage are desired.
>
> 2. In figure 2, there are cases where test label distribution is different from training and validation distribution, how this affects the results, especially considering PAS that is based on class normalization.
>
> > For this point, we have realized we have misled the readers in our presentation, due to Figure 2. Please, see **Truncated dataset** and **Distribution shift / robustness to imperfect prevalence estimation** in the general response.
>
>
> 3. Can this method be extended to regression problems with class imbalance, where the continuous target variable has a highly skewed distribution with certain value ranges, much more frequent than others?
>
> > That's an interesting idea, thanks for sharing it. This could be a good direction for future work. In a regression setting, one can imagine discretizing the outcomes space. The conditional thresholds would then be computed on each subdivision of the space and then our method INTERP-Q could be directly applied. There is no guarantee that the resulting set will form a contiguous interval (but you could easily fix this by taking the convex hull if desired).
> Adapting PAS/WPAS to this setting is less straightforward as it relies on the idea that the score represents a conditional probability/density. Propositions 1-2 could be adapted to a continuous setting and the optimal set would depend of $f(y|x)/f(y)$ where $f$ is the (conditional) density. Then we can adapt for example the methodology of Lei et al. 2013 (*Distribution free prediction sets*) based on thresholding the KDE of the densities and then weighted the score by the estimated density of $y$.
>
> We hope you are satisfied with this response. Please let us know if you have any further questions!

---

> > ### Comment · Reviewer_mwex · 2025-11-27
> >
> > Thanks to the authors for their response.
> >
> > The proposed methods are supported by solid theoretical foundations and extensive experimental results; however, the conceptual novelty appears somewhat limited. I also note that the approach is mainly designed for softmax-based scores and was not demonstrated over other scoring mechanisms, and it does not address distribution shifts, which may limit its applicability.
> >
> > Overall, I will maintain my original score, and I remain generally positive about acceptance.

---

> > > ### Author Response · Authors · 2025-12-01
> > > **Response to Reviewer mwex**
> > >
> > > We thank the reviewer for their time and for being supportive of our paper's acceptance.
> > >
> > > A brief response on “conceptual novelty”: Consider Angelopoulos et al. 2020 [1]. Although, at its core, all the paper does is propose a new conformal score function called RAPS, it’s been very influential (garnering 500+ citations). The primary reason is not due to its conceptual novelty (although this could certainly be a factor), but rather because it is practical (easily implementable) and works well in practical applications. We see our paper as filling the same sort of gap, but in the long tailed setting.
> > >
> > > [1] Angelopoulos, Anastasios, et al. "Uncertainty sets for image classifiers using conformal prediction." arXiv preprint arXiv:2009.14193 (2020).

---

### Official Review · Reviewer_5apV · 2025-11-01

**Soundness:** 3
**Presentation:** 3
**Contribution:** 2
**Rating:** 6
**Confidence:** 4

**Summary:**

The paper studies conformal prediction for very imbalanced multi class problems where a few classes have many examples and many classes have very few. The goal is to produce prediction sets that keep marginal coverage guarantees while improving coverage for rare classes without exploding set size. The authors offer two main ideas. First, a new score called prevalence adjusted softmax and its weighted version that aim directly at macro coverage, with an option to upweight classes of interest such as endangered species. Second, an interpolation procedure called INTERP Q that blends classwise and standard conformal thresholds to trade off set size and class conditional coverage, with a conservative marginal guarantee. They test on Pl@ntNet 300K and iNaturalist 2018 and report better coverage for tail classes at small or moderate set sizes, plus a study with simple human decision models.

**Strengths:**

Targeting macro coverage with a simple change to the score is a neat idea. It connects the oracle form of the optimal set for macro coverage to a practical score based on p hat of y given x divided by the estimated prevalence. The weighted version lets users push coverage toward special subsets like at risk species.

The paper is easy to follow. The problem is well motivated with plant identification. The two approaches are separated and labeled. Table 1 is a good map of methods and guarantees.

Selective set prediction in long tailed regimes is common in biodiversity and medicine and also in open world recognition. A method that improves tail coverage while keeping sets short is directly useful for real labeling workflows and can slow collapse of rare labels in human in the loop systems.

**Weaknesses:**

PAS relies on p hat of y given x and an estimate of label prevalence. In real systems there is often label shift between train, calibration, and test. The paper does not test robustness under such shift, even though label shift directly changes the p of y term that PAS divides by.

The 1 minus 2 alpha lower bound is likely conservative, as the authors note, but the paper does not quantify the realized marginal coverage gap across settings or give a simple correction to hit a target level.

Most results use softmax scores from a standard ResNet trained with cross entropy, with one mention of focal loss in the appendix. Since the approach is driven by score quality, the work would benefit from a broader check across stronger long tail learners such as logit adjusted training and from alternative scores such as APS or label ranking scores, even if set sizes grow

The human models are an expert verifier and a random guesser, plus mixtures. That is a useful first look, but real users are neither.

**Questions:**

How sensitive is PAS to misspecified prevalence. If the true p of y differs from the training estimate, can you still expect macro coverage gains.

Can you provide a practical scheme to choose tau on the calibration fold to meet a specific marginal coverage while improving class conditional coverage.

Have you tried teaching a small correctness predictor on the calibration set and using that as the score inside PAS or as a rank corrector.

Macro coverage can look good while a handful of classes are still far below target. Can you add plots of the full distribution of per class coverage, not just the fraction below fifty percent.

For a practitioner who wants sets of average size at most three while lifting average coverage of at risk species above a target, can you give a small recipe that picks alpha and lambda and, for INTERP Q, tau.

---

> ### Author Response · Authors · 2025-11-21
> **Addressing weaknesses**
>
> Thank you for your positive review. We have updated our draft to address some of the points you raised.
>
> 1. PAS relies on p hat of y given x and an estimate of label prevalence. In real systems there is often label shift between train, calibration, and test. The paper does not test robustness under such shift, even though label shift directly changes the p of y term that PAS divides by.
> >Please see the **Distribution shift / robustness to imperfect prevalence estimation** part of our general response.
>
> 2. The 1 minus 2 alpha lower bound is likely conservative, as the authors note, but the paper does not quantify the realized marginal coverage gap across settings or give a simple correction to hit a target level.
> > In our experiments we compute the marginal coverage of all methods, including INTERP-Q (see the rightmost column on Figure 3). We observe that INTERP-Q does not have marginal coverage worse than roughly $1-\alpha-0.01$. If a user wants a theoretical guarantee of $1-\tilde \alpha$ coverage, they can simply run INTERP-Q at level $\alpha = \tilde \alpha /2$.
>
> 3. Most results use softmax scores from a standard ResNet trained with cross entropy, with one mention of focal loss in the appendix. Since the approach is driven by score quality, the work would benefit from a broader check across stronger long tail learners such as logit adjusted training and from alternative scores such as APS or label ranking scores, even if set sizes grow
>
> > We chose to focus on cross entropy because it is generally the first loss that people try for classification tasks, even in long-tailed settings (see e.g., Pl@ntNet-300K baselines provided in [1]). Furthermore, after running a robustness check using focal loss, we found that the relative performance of our methods relative to baselines was preserved.
>
> 4. The human models are an expert verifier and a random guesser, plus mixtures. That is a useful first look, but real users are neither.
>
> > We agree! It is our hope that our simple attempt at quantifying the joint effect of coverage and set size on decision quality will spur future research in this direction.

---

> ### Author Response · Authors · 2025-11-21
> **Addressing questions**
>
> 1. How sensitive is PAS to misspecified prevalence. If the true p of y differs from the training estimate, can you still expect macro coverage gains.
>
> >Please see the **Distribution shift / robustness to imperfect prevalence estimation** part of our general response.
>
> 2. Can you provide a practical scheme to choose tau on the calibration fold to meet a specific marginal coverage while improving class conditional coverage.
>
> > To get guaranteed $1-\alpha$ coverage, you simply need to run INTERP-Q at the $\alpha/2$ level. The parameter $\tau$ could then be chosen to reach an appropriate set size/macro-coverage trade off decided by the user. If only the set size matters, $\tau$ should be set to $0$ (standard CP), at the opposite end, $\tau =1$ gives classwise CP. The kneedle technique proposed by [2] could also be a way to automatically select a good trade off between these two objectives.
>
> 3. Have you tried teaching a small correctness predictor on the calibration set and using that as the score inside PAS or as a rank corrector.
>
> > Could you clarify what you mean by this?
>
> 4. Macro coverage can look good while a handful of classes are still far below target. Can you add plots of the full distribution of per class coverage, not just the fraction below fifty percent.
>
> > Thank you for this remark. In fact, this information is already conveyed in the decision accuracy plots (Figure 5, expert proportion = 100%) but can be easily missed. We have now emphasized in the paper that the decision accuracy of an expert corresponds to the class-conditional coverage. We observe that PAS improves the coverage of the classes with the lowest class-conditional coverage.
>
> 5. For a practitioner who wants sets of average size at most three while lifting average coverage of at risk species above a target, can you give a small recipe that picks alpha and lambda and, for INTERP Q, tau.
>
> > In practice, it could be interesting to have the reverse point of view: fixing an average size of interest for the prediction set and setting $\alpha$ and/or $\tau$ to reach it. This is in the spirit of set valued classification (see e.g. [3] for an overview), but conformal prediction works focus primarily the coverage. Note also that, in general, it is impossible to target simultaneously a fixed average set size and a coverage. It is always a trade off between these two quantities and you can only fix one quantity and trying to minimize (or maximize) the other.
> >
> > In CP, the choice is to fix the coverage. Nevertheless, if a user targets a specific set size, a grid search can be made over $\alpha$ and $\lambda$/$\tau$ to get it. Since for $\alpha =1$, all sets are empty, a solution must exist but possibly with a poor guarantee if $\alpha$ is large. We recommend selecting a solution with the smallest possible $\alpha$.
>
> >  [1] Garcin et al., Pl@ntNet-300K: a plant image dataset with high label ambiguity and a long-tailed distribution, NeurIPS 2021.
>
> >  [2] Satopää et al., Finding a “Kneedle” in a Haystack: Detecting Knee Points in System Behavior, 2011.
>
> >  [3] Chzhen et al., Set-valued classification--overview via a unified framework, 2021.
>
> Please let us know if you have any further questions.

---

### Official Review · Reviewer_pAFA · 2025-11-01

**Soundness:** 2
**Presentation:** 3
**Contribution:** 2
**Rating:** 6
**Confidence:** 3

**Summary:**

The paper applies conformal prediction (CP) to an extremely long-tailed classification setting. The standard CP tends to ensure coverage of common classes and under/miss coverage of the rare ones, whereas classwise CP covers each class but produces unusable prediction sets. The authors address this challenge by 2 approaches: a) PAS / WPAS -> the idea here is to, instead of trusting the raw model probabilities, recalibrate label scores by their rarity, so that rare classes get a boost. The authors claim this results in smaller prediction sets compared to classwise CP, and is also fairer to rare classes compared to standard CP. b) INTERP-Q -> instead of score recalibration, this approach uses a controllable parameter that provides a cutoff threshold that balances between standard CP and classwise CP thresholds. It gives the user a controllable knob that can decide how much to protect tail classes versus how large a prediction set can be.
Empirically, they compare against standard baselines and the evidence supports their claims.

**Strengths:**

Strength

a) The main paper is well motivated, mostly clear and easy to follow.

b) It tackles conformal prediction in the extreme long-tailed scenario, which is practically important.

c) The class coverage vs prediction set size tradeoff as a problem formulation itself seems novel. The two proposed approaches also appear reasonably original.

d) Empirical studies are convincing, and their human decision maker simulation experiment seems interesting to me.

**Weaknesses:**

Weaknesses

a) I could understand the working of PAS/WPAS and INTERP-Q, but I couldn't clearly find the motivation for utilizing either/or both of them. The two methods seem to address different parts of the pipeline, but the paper does not clearly explain a practical guideline when a practitioner should pick PAS/WPAS, INTERP-Q, or use them together.

b) The experimental details in the appendix mention utilizing a truncated version with n-core filtering with n = 101. I am curious: doesn't this contradict the problem the authors are trying to solve? The paper is motivated by the extreme long tail, but the analysis is done only on this truncated subset. It would be helpful to clarify whether the main conclusions still hold for the truly rare classes that were filtered out.

c) Based on my understanding, PAS / WPAS is motivated by the hypothesis that reweighting by class prevalence is optimal for trading off average per-class coverage and set size. However, in the implementation, PAS simply recalibrates standard split CP scores, which only guarantees marginal coverage, not class-conditional coverage. So the claim that there are fewer under-covered long-tailed classes seems to be empirical rather than a strict guarantee to me. Similarly, INTERP-Q appears to be only loosely bound to the standard finite-sample marginal guarantee. Neither approach provides a strict class-conditional guarantee, which makes the strength of claims less clear.

d) The paper only evaluates using splits where calibration and test share the same long-tailed label frequencies. It would be interesting to see whether the proposed methods hold up when the test distribution differs from calibration.  I believe stress-testing robustness to even basic shifts would better support the practicality claims of the paper.

e) Comparison with standard CP, classwise CP, and the clustered variant is good. But I would like to see how the method fares against some recent methods that seem to have similar motivation/methodologies [1][2].


[1] Liu, Shuqi & Huang, Jianguo & Ong, Luke. (2025). Conformal Prediction Meets Long-tail Classification.

[2] Yuanjie Shi, Subhankar Ghosh, Taha Belkhouja, Janardhan Rao Doppa, and Yan Yan. (2024). Conformal prediction for class-wise coverage via augmented label rank calibration (RC3P). NeurIPS 2024.

**Questions:**

Please refer weakness.

---

> ### Author Response · Authors · 2025-11-21
> **Addressing weaknesses**
>
> Thank you for your review and helpful feedback. We have updated our draft based on the points you raised to further strengthen our paper.
>
>
> a. *I could understand the working of PAS/WPAS and INTERP-Q, but I couldn't clearly find the motivation for utilizing either/or both of them. The two methods seem to address different parts of the pipeline, but the paper does not clearly explain a practical guideline when a practitioner should pick PAS/WPAS, INTERP-Q, or use them together.*
>
> > We believe this is addressed in Section 3.1:
> > "(iv) STANDARD with PAS is Pareto optimal, in the sense that at any marginal coverage level, there is no method that simultaneously achieves better set size and class-conditional/macro- coverage. This suggests that STANDARD with PAS is a good starting place for practitioners due to its simplicity and strong performance on all metrics. However, INTERP-Q is also of practical value since its tunable parameter allows us to choose where we want to be on the trade-off curve between set size and class-conditional coverage without significantly changing the marginal coverage. Note that the two methods can also be combined, as presented in Appendix D.2, Figure 9."
> > Please let us know if there are specific points that you feel are not addressed.
>
> b. *The experimental details in the appendix mention utilizing a truncated version with n-core filtering with n = 101. I am curious: doesn't this contradict the problem the authors are trying to solve? The paper is motivated by the extreme long tail, but the analysis is done only on this truncated subset. It would be helpful to clarify whether the main conclusions still hold for the truly rare classes that were filtered out.*
> > Please see the general response regarding **Truncated dataset** about this issue.
>
> c. *Based on my understanding, PAS / WPAS is motivated by the hypothesis that reweighting by class prevalence is optimal for trading off average per-class coverage and set size. However, in the implementation, PAS simply recalibrates standard split CP scores, which only guarantees marginal coverage, not class-conditional coverage. So the claim that there are fewer under-covered long-tailed classes seems to be empirical rather than a strict guarantee to me. Similarly, INTERP-Q appears to be only loosely bound to the standard finite-sample marginal guarantee. Neither approach provides a strict class-conditional guarantee, which makes the strength of claims less clear.*
> > See general response **PAS coverage guarantees**. We do not target class conditional guarantee but rather a trade-off between class-conditional coverage and set size while maintaining marginal coverage.  Targeting class-conditional guarantees uniformly for all classes is unpractical in long tail settings: as illustrated in our experiments (Classwise CP), this would lead to huge sets (almost all the classes), making them useless for practitioners.
>
> d. *The paper only evaluates using splits where calibration and test share the same long-tailed label frequencies. It would be interesting to see whether the proposed methods hold up when the test distribution differs from calibration. I believe stress-testing robustness to even basic shifts would better support the practicality claims of the paper.*
>
> > See general response **Distribution shift / robustness to imperfect prevalence estimation**.
>
> e. *Comparison with standard CP, classwise CP, and the clustered variant is good. But I would like to see how the method fares against some recent methods that seem to have similar motivation/methodologies [1][2].*
>
> > [1] This paper is contemporaneous to ours, as it became available a month before the ICLR submission on arXiv (https://arxiv.org/abs/2508.11345). As the code is not available online and the paper is not published yet, we have not considered it as a baseline.
>
> >[2] We are working on implementing this baseline. For more details, see general response **Conformal scores and competing methods**.

---

### Official Review · Reviewer_mYkq · 2025-11-08

**Soundness:** 3
**Presentation:** 2
**Contribution:** 2
**Rating:** 6
**Confidence:** 3

**Summary:**

Existing conformal prediction methods often trade off between poor class-conditional coverage and overly large prediction sets. This paper proposes a new non-conformity score, PAS (and its weighted variant WPAS), which aims to interpolate between marginal and class-conditional conformal prediction. The goal is to achieve better coverage–efficiency balance under long-tailed label distributions.

**Strengths:**

1.	Addresses the practically relevant problem of conformal prediction under long-tailed label distributions.

2.	The proposed PAS/WPAS scores are simple and easy to implement.

3.	The experiments offer preliminary evidence that the proposed method improves coverage fairness under long tailed setting.

**Weaknesses:**

1.	Truncated dataset setup: The test datasets are balanced by truncating rare classes and retaining those with more than 100 samples per class. The choice of this threshold is not explained. Why 100? In more realistic scenarios where both calibration and test sets are long tailed, how would the proposed method perform?

2.	Motivation: The new non-conformity score (PAS) is motivated by an oracle analysis showing that the optimal set depends on p(y|x)/p(y), but this only characterizes an ideal solution rather than a provably better practical formulation. Overall, the motivation appears ad-hoc, and it remains unclear what concrete limitation of existing scores (e.g., APS, LAC) PAS resolves under long-tailed distributions.

3.	Bassline: The paper does not include [1], which also targets class-conditional coverage under imbalanced settings, which appears highly relevant to this work. In addition, standard non-conformity scores such as APS [2] and LAC [3] are omitted.

4.	Score comparison: Since APS emphasizes X-conditional coverage and LAC minimizes the expected set size, including them would provide a more informative comparison of the adaptiveness–efficiency tradeoff that PAS/WPAS aims to improve. It would also be helpful to clarify whether a weighted combination of APS and LAC could already achieve a similar trade-off, and how PAS/INTERP-Q compares in that respect.

5.	Evaluation Metric: The study reports FracBelow50% and UCG but omits the more intuitive Under-Coverage Ratio (UCR) used in [1], which reflects the fraction of classes that fail to meet the target coverage and would provide a more informative evaluation.

[1] Yuanjie Shi, Subhankar Ghosh, Taha Belkhouja, Jana Doppa, and Yan Yan. Conformal prediction for class-wise coverage via augmented label rank calibration. NeurIPS 2024.

[2] Yaniv Romano, Matteo Sesia, and Emmanuel J. Candes. Classification with valid and adaptive coverage. NeurIPS, 2020.

[3] Mauricio Sadinle, Jing Lei, and Larry Wasserman. Least ambiguous set-valued classifiers with bounded error levels. J. Amer. Statist. Assoc 2019.

**Questions:**

Please see the weaknesses.

---

> ### Author Response · Authors · 2025-11-21
> **Addressing weaknesses**
>
> Thank you for your thoughtful feedback and for highlighting that our method tackles a practically important problem with easily implementable methods. In our revised draft, we have incorporated your feedback to further strengthen our paper.
>
> 1. Truncated dataset setup: The test datasets are balanced by truncating rare classes and retaining those with more than 100 samples per class. The choice of this threshold is not explained. Why 100? In more realistic scenarios where both calibration and test sets are long tailed, how would the proposed method perform?
> > Thank you for raising this point of confusion. Please see the **Truncated dataset** section of the global response. We would also like to emphasize that our main experiments (see Fig 3) are on untruncated datasets that have very long tails.
>
> 2. Motivation: The new non-conformity score (PAS) is motivated by an oracle analysis showing that the optimal set depends on p(y|x)/p(y), but this only characterizes an ideal solution rather than a provably better practical formulation. Overall, the motivation appears ad-hoc, and it remains unclear what concrete limitation of existing scores (e.g., APS, LAC) PAS resolves under long-tailed distributions.
>
> > LAC/softmax targets the optimal set under marginal coverage constraint. Applying classwise conformal prediction with it (Classwise in the paper) targets the optimal under classwise coverage constraint. Our score PAS targets macro coverage to interpolate between the large size of Classwise and the poor conditional coverage of Standard. A striking example in Figure 3 is for $\alpha=0.2$ and Pl@ntNet dataset, PAS obtains a macro-coverage of 0.75 against almost 0.3 for Standard with an increase of average size of 2 (over 1000 classes). On the other hand Classwise sets contain almost all the classes. We have clarified this objective in the new version of paper.
> > As discussed in **Conformal scores and competing methods** of our global response, APS targets a X-conditional coverage constraint, which is not our objective. Its poor effectiveness in the long tailed setting has been exhibited by Ding et al. (2023) which has lead us to not consider it.
>
> 3. Baseline: The paper does not include [1], which also targets class-conditional coverage under imbalanced settings, which appears highly relevant to this work. In addition, standard non-conformity scores such as APS [2] and LAC [3] are omitted.
>
> > Please see **Conformal scores and competing methods** of our global response.
>
> 4. Score comparison: Since APS emphasizes X-conditional coverage and LAC minimizes the expected set size, including them would provide a more informative comparison of the adaptiveness–efficiency tradeoff that PAS/WPAS aims to improve. It would also be helpful to clarify whether a weighted combination of APS and LAC could already achieve a similar trade-off, and how PAS/INTERP-Q compares in that respect.
>
> > Please see **Conformal scores and competing methods** of our global response.
>
> 5. Evaluation Metric: The study reports FracBelow50% and UCG but omits the more intuitive Under-Coverage Ratio (UCR) used in [1], which reflects the fraction of classes that fail to meet the target coverage and would provide a more informative evaluation.
>
> > The full distribution of per class coverage is actually already conveyed in the decision accuracy plots (Figure 5, expert proportion = 100%), which provides more precise information than the UCR. We see that PAS increases the coverage of the most poorly covered classes compared to using the softmax score.
>
> Please let us know if you have any remaining questions!

---

### Author Response · Authors · 2025-11-21
**Global response (pt. 1)**

Here, we provide some information that is worth sharing with all reviewers. We address more specific comments in individual responses to each reviewer.

**Truncated dataset** (Reviewer mYkq, pAFA): _tl;dr: We created the truncated datasets to ensure that we can properly evaluate class-conditional metrics._
In an ideal world where we had infinitely large test sets for Pl@ntNet-300K and iNaturalist, there would be no need to create truncated versions. Unfortunately, we have finite test sets, and this creates a challenge when evaluating _class-conditional_ performance of the various methods. Due to the long-tailed distribution, there are many classes for which we have only a handful of test examples (as low as 1). If we ask for coverage of 0.90 but we only have one test example, it is difficult to tell if we are achieving the correct coverage, as the empirical coverage computed on one test example is either 1 or 0. To ensure that we have enough data to compute class-conditional metrics, we create truncated versions where each class has 100 test examples (this 100 threshold was chosen somewhat arbitrarily, but felt to us to be enough to allow for good estimation). However (and we admit this is not well-explained in the original main paper), we still assume that the ultimate test distribution we care about is the same as the train/val distribution. So to compute marginal metrics using the balanced test set, we compute the class-conditional metric and take a weighted average according to the class prevalences in the train set. When we recreate Figure 3 on the truncated datasets (see Figure 7 of the Appendix), we get results that are very similar to the full dataset evaluation, so we defer it to the Appendix. However, in the simulated decision-maker experiments in Section 3.3, we find it necessary to use the truncated dataset because we are no longer reporting aggregations of class-conditional metrics (as in Figure 3) but rather the class-conditional metrics themselves. We have added text to our paper to better describe this.

**Distribution shift / robustness to imperfect prevalence estimation** (Reviewer pAFA, 5apV, mwex):
 * Conformal prediction methods fundamentally rely on the assumption that calibration and test data share the same distribution (or are exchangeable). Developing conformal approaches that can accommodate distribution shifts is an important challenge, but a separate objective from the one addressed in this paper. Some work has been done on label shift in conformal prediction settings, for instance, Podkopaev and Ramdas (2021), but we believe it falls outside the scope of our current work.
    Furthermore, distribution shift is not a significant problem in the citizen science applications we are motivated by. The data collected by the user, which is used for model training and calibration, and the test examples the model is deployed on come from the same (citizen driven) data distribution.
* What if we don’t perfectly know the p(y) of the test distribution?
>    * Running PAS with an imperfect estimate $\hat{p}(y)$ can be viewed as running WPAS where the weight on class y is $w(y) = p(y)/\hat{p}(y)$. In other words, running PAS with $\hat{p}$ approximates the sets that achieve the optimal trade off between the $p(y)/\hat{p}(y)$-weighted macro-coverage and set size (while achieving the desired marginal coverage guarantee). As $\hat{p}(y)$ gets closer to $p(y)$, PAS will more closely approximate the sets that optimally trade off _unweighted_ macro-coverage and set size.
>    * In our experiments, since we know that the train, calibration, and test datasets are all from the same distribution, we estimate $\hat{p}$ on the train data, since it is more plentiful than the calibration data. If the train distribution does not match the test distribution, we can instead estimate $\hat{p}$ on the calibration dataset (which, under the standard conformal prediction assumption, is the same as the test distribution)

 **PAS coverage guarantees:** Our key insight is that to optimally trade off set size and macro-coverage (average class-conditional coverage), we should threshold on p(y|x) / p(y). Two ways of setting the threshold is to achieve $1-\alpha$ macro-coverage or to achieve $1-\alpha$ marginal coverage. These two thresholds are different in general. We choose to set the threshold to achieve $1-\alpha$ marginal coverage, as marginal coverage is an important desiderata in many settings (and is a goal of almost all conformal prediction methods). We have updated our paper to emphasize that PAS is not designed to create prediction sets with $1-\alpha$ macro-coverage.

---

### Author Response · Authors · 2025-11-21
**Global response (pt. 2)**

**Conformal scores and competing methods**:
 * LAC score (for Reviewer mYkq): we do compare against the LAC score, but we refer to it as the *softmax* score. In the literature, both LAC and softmax are used as names for this score. We have added a note in our paper to highlight that what we call the softmax score is also known as LAC
 * APS/RAPS/SAPS score : It has been observed in several works on many-class classification settings that APS and its variants produce larger sets than softmax (e.g., Ding et al., 2023). The main reason to use APS is to get approximate X-conditional coverage. We assume that this is not an important goal in our setting.
 * Ding et al. (2023)'s method (for Reviewer mwex): this method is called Clustered in our experiments (see for instance Fig. 3). It does not perform well in long-tailed settings, as it defaults to Standard CP for the many classes in the tail.
 * RC3P method by Shi et al. (2024): We are currently working on implementing this baseline and hope to have results in the coming days. However, we expect this method to suffer from similar problems as Classwise, as it similarly has to estimate parameters (a score and rank threshold) for each class.

**Hyperparameter selection:**
 * [WPAS parameter] $\lambda$ (at-risk species weight) should be chosen based on how much more you value a correctly identified instance of an at-risk species than a not-at-risk species. For instance, if it is 10x more valuable to correctly identify an at-risk species, then set $\lambda=10$. This depends on the application.
 * [INTERP-Q parameter] For the interpolation parameter $\tau$, choosing $\tau=0.75$ seems to provide a good balance in our experiments. More generally, this parameter could be tuned in a more systematic way, possibly on left out data in the same manner used to tune hyperparameters in other conformal methods such as RAPS. This held out valuation can be combined with the kneedle technique from [1] to identify the best trade off point. To get a theoretical guarantee of $1-\alpha$ coverage, INTERP-Q can be at the $\alpha/2$ level.

**Summary of changes to paper**. We describe the changes we have made to incorporate reviewer feedback. These changes are highlighted in **red** in the paper pdf.
* **Tightness of INTERP-Q coverage bound**: One reviewer questioned the tightness of the $1-2\alpha$ coverage guarantee of INTERP-Q. We have added an example of distribution for which this lower bound is almost attained (up to $\alpha^2$). See the updated commentary following Proposition 3 and the corresponding formal statement in Appendix B.3.
* **Better explanation of truncated dataset**: We have updated the discussion about the truncated dataset to clarify the purpose of its construction: having correct estimates of class-conditional coverages.
* **PAS guarantee**, we have emphasized that PAS does not target directly a macro coverage guarantee but better handles the trade-off of macro-coverage and set size. In particular, we have clarified the objective or our methods and replaced $1-\alpha$ by $\beta$ in the optimization problem (8) and Proposition 1 and 2 as it seems to mislead readers.
* **Improved evaluation visualization**: We added a $y=1-\alpha$ line in Figure 5 to show when the decision accuraries (which equals the classwise coverage for $\gamma=100\%$) goes below this targeted level (linked to Reviewer mYkq's "Evaluation Metric" remark)
* Other minor edits have been added throughout the paper to clarify various points.

[1] Satopää et al., Finding a “Kneedle” in a Haystack: Detecting Knee Points in System Behavior, 2011.

---

### Author Response · Authors · 2025-11-25
**Results for additional baseline (RC3P)**

**RC3P results added**: Per the suggestion of some reviewers, we have implemented the RC3P method from Shi et al., 2024. The results can be seen in the **updated Figure 7** (p. 25) that is available in the newly uploaded draft. As we expected, this method does not perform well in the extremely long-tailed setting that is considered in our paper. This is firstly because RCP3 is based on Classwise CP and strongly relies on the estimation of the conditional quantile, which can be very bad for classes with few calibration examples. Secondly, RCP3 is based on the estimation of the conditional top-k error for all classes, which is also hard when classes have few calibration examples. Moreover, as these quantities are estimated on the calibration set, the exchangeability assumption is violated which explains the marginal coverages below $1-\alpha$. The methods we propose strictly dominate RC3P; for a given average set size, we get much better class-conditional coverage.
We have added this discussion in Appendix D.1.

---

### Meta-Review · Area_Chair_euBp · 2026-01-10

**Summary:**

This paper considers the problem of conformal prediction based uncertainty quantification in the long-tailed classification settings (tail refers the distribution of #training examples per class). Towards the goal of producing smaller uncertainty sets by retaining marginal coverage guarantee and improving the coverage for rare classes, the paper provides two ideas: 1) A new scoring function based on adjusted softmax and its weighted version, and 2) an interpolation approach to integrate classwise and marginal CP thresholds to trade-off prediction set size and class-conditional coverage. Experiments demonstrate the effficacy of the proposed approaches. The paper is extremely well-written and easy to follow, and the AC appreciates the authors' for it!

All reviewers were generally positive about the paper and ideas, but had questions including experimental comparison with recent methods in similar settings and hyper-parameters, tightness of coverage bound and related guarantees, explanation of truncated dataset, and contextualization with the broader CP literature. Authors' have answered most of these questions satisfactorily and revised the paper to reflect the changes.

Overall, I'm quite confident that all the reviewers' would have recommended accepting the paper based on the author rebuttal. Therefore, I recommend accepting this paper as poster as the overall contribution doesn't rise to the level of a spotlight.

**Reviewer Concerns:**

Based on my reading of the author rebuttal and the revised paper, almost all the concerns were addressed satisfactorily.

**Reviewer Scores:**

Three out of four reviewers would have likely increased the score from 6 to 7 (if allowed), but mostly won't rise to 8.

The fourth reviewer kept the score same (6).

---

### Decision · Program_Chairs · 2026-01-26

Accept (Poster)